# Underestimation of denitrification rates from field application of the $^{15}$N gas flux method and its correction by gas diffusion modelling

Reinhard Well[1], Martin Maier[2], Dominika Lewicka-Szczebak[1], Jan-Reent Köster[1]; and Nicolas Ruoss[1]

[1]Thünen Institute, Climate-Smart Agriculture, Braunschweig, Germany
[2]Forest Research Institute Baden-Württemberg, Dep. Soil and Environment, Freiburg, Germany

*Correspondence to*: Reinhard Well (reinhard.well@thuenen.de)

**Abstract.** Common methods for measuring soil denitrification *in situ* include monitoring the accumulation of $^{15}$N-labelled $N_2$ and $N_2O$ evolved from $^{15}$N-labelled soil nitrate pool in closed chambers that are placed on the soil surface. Gas diffusion is considered to be the main transport process in the soil. Because accumulation of gases within the chamber decreases

concentration gradients between soil and chamber over time, the surface efflux of gases decreases as well and gas production rates are underestimated if calculated from chamber concentrations without consideration of this mechanism. Moreover, concentration gradients to the non-labelled subsoil exist, inevitably causing downward diffusion of $^{15}$N-labelled denitrification products. A numerical 3-D model for simulating gas diffusion in soil was used in order to determine the significance of this source of error. Results show that subsoil diffusion of $^{15}$N-labelled $N_2$ and $N_2O$ - and thus potential underestimation of

denitrification derived from chamber fluxes - increases with chamber deployment time as well as with increasing soil gas diffusivity. Simulations based on the range of typical soil gas diffusivities of unsaturated soils showed that the fraction of $N_2$ and $N_2O$ evolved from $^{15}$N-labelled $NO_3^-$ that is not emitted at the soil surface during one hour chamber closing is always significant with values up to >50 % of total production. This is due to accumulation in the pore space of the $^{15}$N-labelled soil and diffusive flux to the unlabelled subsoil. Empirical coefficients to calculate denitrification from surface fluxes were derived

by modelling multiple scenarios with varying soil water content. Modelling several theoretical experimental set-ups showed that the fraction of produced gases that are retained in soil can be lowered by lowering the depth of $^{15}$N-labelling and/or increasing the length of the confining cylinder.

Field experiments with arable silt loam soil for measuring denitrification with the $^{15}$N gas flux method were conducted to obtain direct evidence for the incomplete surface emission of gaseous denitrification products. We compared surface fluxes of

25 $^{15}$N$_2$ and $^{15}$N$_2O$ from $^{15}$N–labelled micro-plots confined by cylinders using the closed chamber method with cylinders open or closed at the bottom, finding 37% higher surface fluxes with bottom closed. Modeling fluxes of this experiment confirmed this effect, however with a higher increase in surface flux of 89%.

From our model and experimental results we conclude that field surface fluxes of $^{15}$N-labelled $N_2$ and $N_2O$ severely underestimate denitrification rates if calculated from chamber accumulation only. The extent of this underestimation increases

with closure time. Underestimation also occurs during laboratory incubations in closed systems due to pore space accumulation

of $^{15}$N-labelled $N_2$ and $N_2O$. Due to this bias in past denitrification measurements, denitrification in soils might be more relevant than assumed to date.

Corrected denitrification rates can be obtained by estimating subsurface flux and storage with our model. The observed deviation between experimental and modeled subsurface flux revealed the need for refined model evaluation which must include assessment of the spatial variability in diffusivity and production and the spatial dimension of the chamber.

.

## 1 Introduction

$N_2O$ reduction to $N_2$ is the last step of microbial denitrification, i.e. anoxic reduction of nitrate ($NO_3^-$) to $N_2$ with the intermediates $NO_2^-$, $NO$ and $N_2O$ (Mueller and Clough, 2014). Commonly applied analytical techniques enable us to quantitatively analyse only the intermediate product of this process, $N_2O$, but not the final product, $N_2$. The challenge to quantify denitrification rates is largely due to the difficulty in measuring $N_2$ production due to its spatial and temporal heterogeneity and the high $N_2$-background of the atmosphere (Groffman et al., 2006). There are three principles to overcome the latter problem: (i) adding $NO_3^-$ highly enriched in $^{15}$N and monitoring $^{15}$N labelled denitrification products ($^{15}$N gas flux method) (e.g. Siegel et al., 1982) ; (ii) adding acetylene to block $N_2O$ reductase quantitatively and estimating total denitrification from $N_2O$ production (acetylene inhibition technique, e.g. Felber et al., 2012) ; (iii) measuring denitrification gases during incubation of soils in absence of atmospheric $N_2$ using gas tight containers and artificial Helium/Oxygen atmosphere (HeO$_2$ method; Scholefield et al., 1997; Butterbach-Bahl et al., 2002). Each of these methods to quantify denitrification rates in soils has various limitations with respect to potential analytical bias, applicability at different experimental scales and the necessity of expensive instrumentation that is not available for routine studies. Today the acetylene inhibition technique is considered unsuitable to quantify $N_2$ fluxes under natural atmosphere, since its main limitation (among several others, e.g. Saggar et al., 2013) is the catalytic decomposition of $NO$ in presence of $O_2$ (Bollmann and Conrad, 1997a, b). This results in unpredictable underestimation of gross $N_2O$ production (Nadeem et al., 2013). The $^{15}$N gas flux method requires homogenous $^{15}$N-labelling of the soil (Mulvaney and Vandenheuvel, 1988). Moreover, under natural atmosphere this method is not sensitive enough to detect small $N_2$ fluxes (Lewicka-Szczebak et al., 2013). Direct measurement of $N_2$ fluxes using the HeO$_2$ method is not subject to the problems associated with $^{15}$N-based methods (Butterbach-Bahl et al., 2013), but the need for sophisticated gas tight incubation systems limits its use to laboratory incubations only. Consequently, the $^{15}$N gas flux method is the only method potentially applicable in field conditions.

Denitrification in ecosystems is complexly controlled by interaction of labile C, abundance and community structure of denitrifiers, pore structure, soil and root respiration, and mineral N dynamics (Müller & Clough, 2014). It is difficult to keep conditions in the lab identical to the field where some conditions dynamically change due to climatic factors, but especially

due to the activity of plants. Hence, field measurements are indispensable for reliable determination of denitrification in ecosystems.

When chamber methods are used to determine soil gas fluxes to the atmosphere, a certain fraction of the produced gas is not emitted at the surface but remains in the soil (Parkin et al, 2011). This is because the accumulation of gases in the closed chamber decreases concentration gradients between soil and chamber atmosphere causing lowering of surface fluxes with increasing chamber deployment time (Healy et al., 1996). This effect has been addressed in numerous studies (Venterea et al., 2009, Healy et al. 1996, Sahoo et al., 2010). To correct bias from this effect, several approaches have been developed and compared (Parkin et al., 2011). Denitrification estimates based on measurements of $N_2$ and $N_2O$ surface fluxes could also be biased by this effect. This had been suggested for the acetylene inhibition technique in the field (Mahmood et al., 1997) and also for the $^{15}N$ gas flux method (Sgouridis et al., 2016). However, to our knowledge the magnitude in possible underestimation of denitrification rates has not been investigated until now. It can be expected that diffusive loss of $^{15}N$-labelled gases to the subsoil is even more relevant than the respective loss of non-labelled soil gases. This is due to the fact that the production of $CO_2$ and trace gases in soil is ubiquitous, whereas the formation of $^{15}N$-labelled denitrification products is limited to the soil volume amended with $^{15}N$-labelled $NO_3^-$.

Estimating bias from the diffusive loss of $^{15}N$-labelled gases could be done by modelling. Previously, denitrification in subsoil had been quantified by fitting measured and modelled steady state concentration of $^{15}N_2 + {}^{15}N_2O$ (Well and Myrold, 2002). Modelling diffusive fluxes of $^{15}N_2 + {}^{15}N_2O$ produced in $^{15}N$-labelled surface soil based on measured surface flux and diffusivity could be used to estimate its accumulation in pore space and diffusive loss to the subsoil. This could be used to quantify denitrification from the sum of surface flux, subsoil flux and storage within the $^{15}N$-labelled soil volume.

Our objectives were thus to determine the significance of the fraction of $^{15}N$-labelled denitrification products produced in $^{15}N$-labelled soil in the field that is not emitted at the soil surface. This was done experimentally and by diffusion modelling. Moreover, we aimed to develop a procedure to determine denitrification rates from surface flux data. We hypothesized that (i) a significant fraction of $^{15}N$-labelled denitrification products is not emitted at the soil surface, (ii) this fraction depends on diffusivity, chamber deployment time and depth of $^{15}N$–labelling, and (iii.) diffusive loss of $^{15}N$-labelled gases to the subsoil is more relevant than accumulation in the pore space of the $^{15}N$-labelled soil.

## 2 Materials and Methods

### 2.1 Principles of the $^{15}N$ gas flux method and gas flux dynamics following $^{15}N$ tracer application

The $^{15}N$ gas flux method consists of quantifying $N_2$ and / or $N_2O$ emitted from $^{15}N$-labelled nitrate applied to soil in order to quantify fluxes from microbial denitrification (Mulvaney, 1988; Stevens et al., 1993) where $N_2$ and $N_2O$ are formed from the

combination of two NO precursor molecules. To quantify denitrification, experimental soil is typically confined by cylinders installed to a certain depth. These micro-plots are amended with $^{15}$N-labelled nitrate either by surface application of the fertilizer (Kulkarni et al., 2011) or by injecting fertilizer solution using needles to achieve homogenous labelling (Sgouridis et al., 2016, Buchen et al., 2016). Emitted $^{15}$N-labelled gases are collected in chambers fitted gas tight on top of the cylinders, typically for periods of one hour or longer. Soil-derived gases mix with background air inside the closed chambers. $N_2$ and $N_2O$ fluxes from the labelled $NO_3^-$ are calculated from the abundance of $N_2$ and $N_2O$ isotopologues (i.e. molecular species that differ in the number of isotopic substitutions (Coplen, 2011)) in the gas accumulating in the chamber.

To measure denitrification in arable soil, depth of confinement, and also of labelling, typically includes the $a_p$ horizon of the soil, i.e. usually depth of tillage. In this horizon, most of denitrification activity is assumed due to its content in soil organic matter, undecomposed plant litter, organic root exsudates, root respiration as well as fertilizer application to the surface (Groffman et al., 2009).

To keep our modelling as simple as possible we assume a simplified process dynamics where in terms of N transformation only nitrate reduction by microbial denitrification occurs with $N_2$ and $N_2O$ as emitted products.

The bias in determining denitrification rates from the accumulation of $^{15}N_2$ and $^{15}N_2O$ is illustrated by a conceptual model (Figs. 1 and 2 a,b). After closing a chamber on top of the $^{15}$N-labelled soil, the timing and magnitude of $^{15}(N_2+N_2O)$ fluxes depend on the chamber volume, denitrification rates of the $^{15}$N-labelled soil and on gas diffusivity within and around this soil (Fig. 1).

We define the fluxes of $^{15}$N- labelled gases as relative fluxes in relation to the production of these gases as follows:

- The *surface flux* is the flux of $^{15}$N-labelled gases to the atmosphere at the soil surface, either into the free atmosphere or into a closed flux chamber. Relative surface flux is the ratio between surface flux rate and production rate.

- Relative *subsoil flux* is the flux rate of $^{15}$N-labelled gases at the lower boundary of the $^{15}$N-labelled soil in relation to the production rate. Subsoil flux occurs always in downward direction and is thus expressed as negative flux.

- Accumulation of $^{15}$N-labelled gases within the $^{15}$N-labelled soil is referred to as *storage flux* which is the increase in the concentration of accumulated $^{15}$N-labelled gases. Relative storage flux is thus storage flux rate in relation to the production rate.

Assuming constancy of denitrification rates and gas diffusivity, the following dynamics in concentration and gaseous fluxes would establish:

- Following $^{15}$N-labelling, production of $^{15}$N–labelled $N_2$ and $N_2O$ would start at constant rates.
- Before closing the chamber, the upper soil boundary is the free atmosphere where gas exchange is fast enough to preclude $^{15}$N accumulation above the soil surface.

- Production leads to accumulation of $^{15}N$-labelled gases and thus to build-up of concentration gradients to the surface and to the subsoil (Fig. 2a), which causes increasing surface and subsoil fluxes while the storage flux decreases (Fig. 2b).

- After a certain time, steady state is reached, where all fluxes reach constancy.

- Closing the chamber changes the upper boundary since chamber concentration increases due to surface flux (Fig. 2a). Consequently, subsoil and storage flux are rising again, whereas surface flux is decreasing.

If diffusivity and volume of $^{15}N$-labelled soil is known and constancy of parameters is long enough to achieve steady state before closing the chamber, then the relative surface flux can be determined by modelling production and diffusion with open chamber until steady state and during the subsequent phase of chamber closing. Production can thus be calculated from modelled relative surface flux and measured surface flux rate.

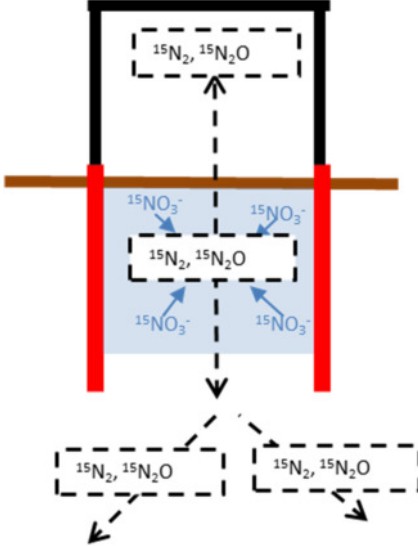

**Fig. 1: Conceptual model describing the dynamics of diffusive fluxes (black dotted arrows) of $^{15}$N-labelled gaseous denitrification products evolved in a $^{15}$N-labelled soil volume (shaded area) that is confined by a cylinder with open bottom to the subsoil, but temporarily closed from above with a flux chamber to collect emitted gases.**

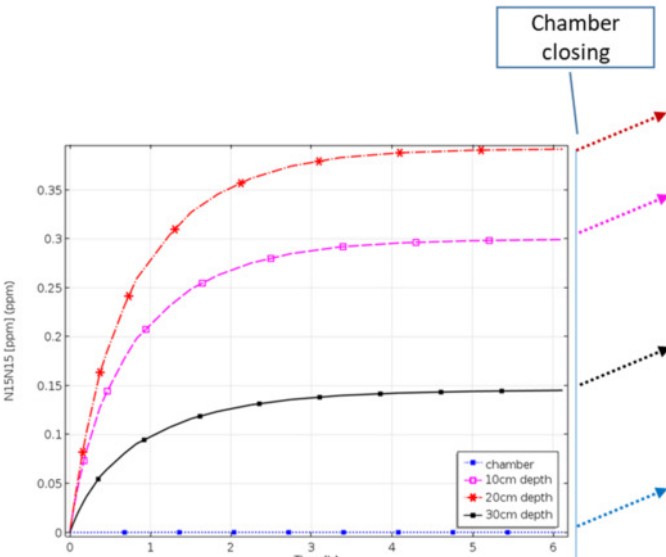

**Figure 2a: Increase in pore space concentrations of N$_2$ evolved from the $^{15}$N-labelled pool after start of denitrification with open chamber when production of $^{15}$N–labelled N$_2$ and N$_2$O would start at constant rates, leading to accumulation of $^{15}$N-labelled gases and thus to build-up of concentration gradients to the surface and to the subsoil. Concentration trends following chamber closure are shown as dotted lines..**

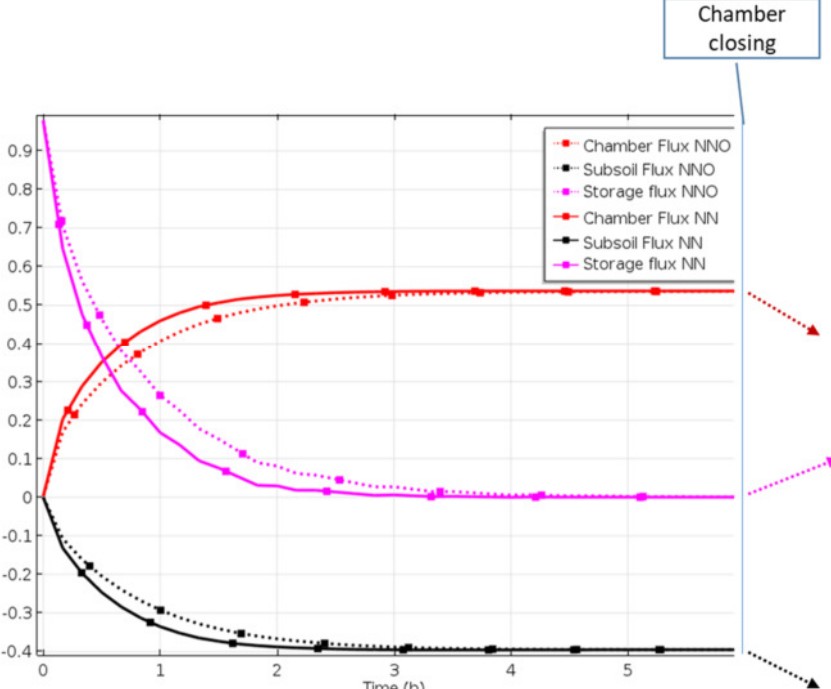

**Figure 2b:** Time course of relative fluxes of $N_2$ and $N_2O$ evolved from the $^{15}N$-labelled pool after start of denitrification with open chamber showing increasing surface and subsoil fluxes while the storage flux decreases until steady state is reached. Trends of fluxes following chamber closure are shown as dotted lines.

## 2.2 Numerical Finite Element Modeling of soil gas transport

### 2.2.1 Conceptual model of the experimental set-up

Numerical finite element modelling (FEM) was used for simulating gas transport during the chamber measurements (COMSOL Multiphysics, Version 5.2 COMSOL Inc., Burlington, Massachusetts, US) to estimate surface and subsurface fluxes of $^{15}N$–labelled $N_2$ and $N_2O$. A conceptual 3D model was built based on geometry and properties of the experimental set-up, this means soil and the cylinder containing the labelled $NO_3^-$, the atmosphere and the chamber. We assumed a soil pore matrix consisting of 2 soil layers with homogenous properties each (total pore volume, soil water content and soil gas diffusivity) into which a gas impermeable cylinder of 15 cm diameter and 35 cm length was vertically installed to a depth of 30 cm. The soil $NO_3^-$ within the cylinder was assumed to be homogenously distributed and labelled with 50 atom % $^{15}N$ to produce homogeneously the isotopologues of $N_2$ and $N_2O$ ($^{14}N^{14}N$, $^{15}N^{14}N$, $^{15}N^{15}N$, $^{14}N^{14}NO$, $^{15}N^{14}NO$, $^{15}N^{15}NO$), while the surrounding soil was not producing any of these gas species. The latter assumption is a simplification to reduce modelling effort, since all of the isotopologues are produced to some extent also from non-labelled N sources outside the $^{15}N$-labelled soil volume. However, due to the high $^{15}N$-enrichment in the labelled soil, the abundance of $^{15}N^{14}N$ is 69 times higher in the

[15]N pool derived fluxes and the abundance of $^{15}N^{15}N$ is even 18,600 times higher, compared to natural abundance of $^{15}N$. Hence, fluxes from non-labelled N pools would not significantly affect the model result. A cylinder shaped gas impermeable cover chamber of totally 20 cm height was used to close temporarily the base cylinder to measure soil gas fluxes. The model assumes that the chamber atmosphere is always homogeneously mixed and that production is constant within the $^{15}N$-labelled soil volume. The cylinder was always assumed open at the bottom unless differently specified.

Two different experimental set-ups were modelled and used in the field. The first experimental set-up (*A_bottom_open*) is described by the conceptual model above and includes an open bottom cylinder containing the labelled $NO_3^-$. In the second set-up (*A_bottom_closed*), the lower end of the cylinder was sealed, so that gases could only be emitted via the surface. This was represented by an additional impermeable thin layer (Table 1).

### 2.2.2 Gas transport modelling

Molecular gas diffusion was assumed to be the only transport mechanism in the soil. The left and right side and the bottom of the modelled domain were defined as impermeable (Neumann boundary condition). The upper boundary of the atmospheric layer was set to atmospheric gas concentrations as Dirichlet boundary condition, and acts therefore as sink for the gases produced. To increase computational efficiency, a 2D axisymmetric modelling approach was chosen since the cylinder and the chamber were round objects. The modelled volume was set to sufficiently large volume with a soil depth of 1.0 m and a diameter of 1.0 m to ensure that the dimension of the modelled area does not affect the modelling outcome within the cylinder and chamber area.

Gas transport was modelled for all isotopologues of $N_2$ and $N_2O$. Diffusivity in free air ($D_0$) was set to 0.193 cm² s$^{-1}$ for $^{14}N^{14}N$ and 0.137 cm² s$^{-1}$ for $^{14}N^{14}NO$ according to Marrero (1972). Diffusivity in free air of $^{15}N^{14}N$ (0.1913 cm² s$^{-1}$), $^{15}N^{15}N$ (0.1896 cm² s$^{-1}$), $^{15}N^{14}NO$ (0.1364 cm²/s) and $^{15}N^{15}NO$ (0.1358 cm²/s) were derived based on their mass following Jost (1960). The relative diffusion coefficient in soil $D_S/D_0$ accounts for the reduced diffusivity in a porous system. $D_S/D_0$ was calculated using the diffusion model of Millington (1959) ($D_S/D_0 = E^2/TP^{2/3}$), where the air-filled pore volume E was calculated as the difference of the total pore volume TP of the soil and the volumetric soil water content SWC, and TP was derived from bulk density. The free atmosphere was assumed to be well mixed single layer and the effective diffusivity was set to $40 \times D_0$, so that the concentration in the atmosphere was kept stable.

### 2.3 Modelling set-ups and scenarios

Different experimental set-ups and scenarios were modelled with the lower end of the cylinder sealed or open (*A_bottom_closed; A_bottom_open*), describing the actually used field set-up (Table 1). Also further theoretical set-ups have been modelled to evaluate the effect of the dimension of the cylinder and labelled zone (B-scenarios).

**Table 1: Modelled set ups**

| Set up | Bottom | Cylinder length | Labelled zone |
|---|---|---|---|
| *A_bottom_open* | Bottom open | 30 cm | 0- 30 cm |
| *A_bottom_closed* | Bottom closed | 30 cm | 0- 30 cm |
| *B_30_30* | Bottom open | 30 cm | 0- 30 cm |
| *B_45_30* | Bottom open | 45 cm | 0- 30 cm |
| *B_45_45* | Bottom open | 45 cm | 0- 45 cm |
| *B_60_45* | Bottom open | 60 cm | 0- 45 cm |

### 2.3.1 Time until steady state after labelling

Time dependent modelling of the open chamber set up was performed to assess the time that is needed after the initiation of the system, i.e. after adding labelled $NO_3^-$, until the production and transport of the $N_2O$ and $N_2$ isotopologues reach a steady state concentration distribution within the soil cylinder and the surrounding soil (Figs. 2a and 2b). This represents the minimum time to be waited after the label application before the first chamber measurement.

### 2.3.2 Modelling chamber measurements

To model chamber measurement, two modelling steps were run. In a first modelling step, steady state concentration distributions assuming steady denitrification were modelled for the open chamber. The resulting concentration distributions were then used in a second modelling step as input for time step 0 for the time dependent modelling of the closed chamber (Figs. 2a and 2b). This approach was used for all modelling scenarios.

### 2.3.3  Modelling the effect of soil moisture

Parameter sweeps were conducted for the set-ups used in the field (*A_bottom_open* and *A_bottom_closed*, Table 1) to assess the theoretical effect of soil moisture, pore volumes and production rates. This was done to account for these transport related effects in the calculation of the flux measurements. For all parameter combinations a new model was calculated.

Total pore volume was set to 0.51 m³ m⁻³ for the parameter sweep which corresponds to a bulk soil density of 1.30g cm⁻³. The soil water contents used for the parameter sweep were 0.2, 0.3, 0.35 and 0.4 m³ m⁻³ and corresponded to a range of $D_S/D_0$ of 0.053-0.210. The production rates of the gas species used for the parameter sweep were chosen (Table 2) so that the outcome of the parameter sweep models covered the range of the observed concentration of the respective species.

**Table 2 Range of parameter values used to assess effect of soil gas transport and production rates**

| Parameter | Parameter range |
|---|---|
| $^{14}N^{14}N$ Production | 3.0-60 nmol m$^{-2}$ s$^{-1}$ |
| $^{15}N^{14}N$ Production | 0.3-6.0 nmol m$^{-2}$ s$^{-1}$ |
| $^{15}N^{15}N$ Production | 0.03-6.0 nmol m$^{-2}$ s$^{-1}$ |
| $^{14}N^{14}NO$ Production | 0.3-6.0 nmol m$^{-2}$ s$^{-1}$ |
| $^{15}N^{14}NO$ Production | 0.03-1.5 nmol m$^{-2}$ s$^{-1}$ |
| $^{15}N^{15}NO$ Production | 0.03-1.5 nmol m$^{-2}$ s$^{-1}$ |
| Soil water Content | 0.2- 0.4 m$^3$ m$^{-3}$ |

The output of the parameter sweeps of scenario *A_bottom_open* and scenario *A_bottom closed* included combinations of soil water content, production rates of the soil core, chamber concentrations, and fluxes into the chamber and into the subsoil of the respective gas species. This dataset allowed for linking the gas concentration in the chamber after 2h at a given soil moisture with the respective production rate. Non-linear functions were fitted to the dataset (PROC NLIN, SAS 9.2, SAS Institute Inc., Cary) so that the original production of a gas species could be directly calculated from the concentration after 2h of the respective gas, the total pore volume and the soil moisture. Instead of soil water content, the soil gas diffusion coefficient $D_S$ was used as factor, which allowed to derive a single functional relationship for all gas species for each scenario. This procedure was chosen as an efficient alternative to inverse modelling of individual datasets as described in Laemmel et al. (2019).

Four additional theoretical experimental set-ups were modelled to assess the effect of the soil cylinder length and the length of the labelled zone within the cylinder (B-scenarios, Table 1). For theses set-ups the same soil parameters as for the field scenarios was used. To assess the effect of soil moisture, the model was run at soil water contents of 0.24, 0.34 and 0.44 m$^3$ m$^{-3}$ .

Underestimation of gas production was quantified as difference between the production ($P_i$) and the mean surface efflux during the chamber closure (mean Efflux$_i$), divided by $P_i$ [underestimation = ($P_i$- mean Efflux$_i$)/$P_i$]. The mean surface efflux during the chamber closure corresponds to a linear approach, e.g.. the flux is calculated using the initial and final gas concentration. Subsoil loss was quantified as mean subsoil flux at the lower end of the core during the chamber closure divided by $P_i$.

### 2.3 Field measurements

Experiments were part of a field campaign to measure $N_2O$ fluxes and denitrification in an arable soil cropped with maize. The soil was a Haplic Luvisol developed in loess (silt loam texture with 83±3% % silt, 15±3% clay, 2±0.5 % sand) with a pH of

6.7±0.1 (in $CaCl_2$), a total organic carbon content of 1.24 ± 0.18 % (TOC) and a total nitrogen content of 0.16±0.02 % N in the 0-30 cm topsoil layer. Experiments were conducted between May 30 and June 4  2016.

Four aluminium cylinders of 35 cm length and tapered at the lower end were driven into the soil to 30 cm depth, thus leaving the upper end 5 cm above the soil surface. [15]N-labelling was conducted May 30 as described previously (Buchen et al., 2016).

Soil columns were fertilized with [15]N-labelled $KNO_3$ (70 atom % [15]N) at 10 mg N $kg^{-1}$, resulting in a fertilizer equivalent of 45 kg N per ha. The tracer was dissolved in distilled water and then applied by injections via 12 equidistant steel capillaries. Defined volumes were injected at 2.5, 7.5, 12.5, 17.5, 22.5 and 27.5 cm depth using a peristaltic pump (Ismatec BVP, Wertheim, Germany) to achieve homogenous labelling at 0 to 30 cm depth. Fluxes of $N_2O$ were determined using the closed chamber method (Hutchinson and Mosier, 1981) with opaque PVC chambers with a volume of 4.42 dm³ (diameter 1.5 dm,

height 2 dm). At each sampling date, chambers were closed and sealed air tight with rubber bands for 120 minutes. Headspace sampling for GC analysis was performed in evacuated screw-cap exetainers (12 mL) in a sampling interval of 0, 20, 40, 60 minutes using a 30 ml syringe. 120 minutes after closing, duplicate headspace samples were taken for GC and IRMS analysis. Flux measurements were conducted daily, but only the final date of this measurement campaign (June 4) was used to evaluate the extent of diffusive loss of [15]N-labelled $N_2$ and $N_2O$ to the subsoil. This was done by comparing conventional flux

measurements with cylinders open to the subsoil or with cylinders closed at the bottom. For the latter, cylinders were carefully removed from the surrounding soil. Soil material extending below the lower end of the cylinders was cut off with a knife. Bottom ends were sealed with plastic foil that was fixed at the outer cylinder wall with adhesive tape. Finally, sealed cylinders were put back to their original position in the surrounding soil in order to keep temperature within the cylinders identical to the surrounding soil. Chambers were fitted on the cylinders again for 120 minutes. Samples were collected from the chambers

as in the conventional flux measurement. Between measurements with open and closed bottom, cylinders remained open at the top for 120 minutes to allow equilibration of soil air with the free atmosphere and thus to release accumulated $^{15}N_2$ and $^{15}N_2O$.

### 2.4 Analysis

**2.4.1 Soil analyses**

Soil water content was determined by weight loss after 24h drying at 110ºC. Soil $NO_3^-$ and $NH_4^+$ were extracted in 0.01 M $CaCl_2$ solution (1:10 ratio) by shaking at room temperature for one hour and $NO_3^-$ and $NH_4^+$ concentrations were determined colorimetrically with an automated analyser (Skalar Analytical B.V., Breda, The Netherlands).

### 2.4.2 Isotopic analysis of $NO_3$

[15]N abundances of $NO_3^-$ ($a_{NO3}$) was measured according to the procedure described in (Eschenbach et al., 2017). $NO_3^-$ was reduced to NO by Vanadium –III- chloride (VCl3). [15]N measurement of produced NO was done with a quadrupole mass spectrometer (GAM 200, InProcess, Bremen, Germany).

### 2.4.3 Total $N_2O$

Samples were analysed using an Agilent 7890A gas chromatograph (Agilent Technologies, Santa Clara, CA, USA) equipped with a pulsed discharge detector (VICI, V-D-3-I-7890-220). Precision, as given by the standard deviation (1σ of four standard gas mixtures) was typically 1.5%.

### 2.4.4.2 Isotopic analysis of $N_2$ and $N_2O$

Gas samples were analysed for m/z 28 ([14]N[14]N), 29 ([14]N[15]N) and 30 ([15]N[15]N) of $N_2$ using a modified GasBench II preparation system coupled to an IRMS (MAT 253, Thermo Fisher Scientific, Bremen, Germany) according to Lewicka-Szczebak et al., (2013). This system allows a simultaneous determination of mass ratios $^{29}R$ (29/28) and $^{30}R$ (30/28) of three separated gas species ($N_2$, $N_2+N_2O$ and $N_2O$), all measured as $N_2$ gas after $N_2O$ reduction in a Cu oven. For each of the analysed gas species, the fraction originating from the [15]N-labelled pool with respect to total N in the gas sample ($f_p$) as well as the [15]N enrichment of the [15]N-labelled N pool ($a_p$) producing $N_2O$ ($a_{p\_N2O}$) or $N_2+N_2O$ ($a_{p\_N2+N2O}$) were calculated after Spott et al. (2006) as described in Lewicka-Szczebak et al. (2017). The residual fraction of $N_2O$ remaining after $N_2O$ reduction to $N_2$ ($r_{N2O}$) is given by the ratio $f_{p\_N2O}/f_{p\_N2+N2O}$. Typical repeatability of $^{29}R$ and $^{30}R$ (1 σ of 3 replicate measurements) was $5\times10^{-7}$ for both values.

### 2.5 Statistics

Results of flux measurements with bottom open or bottom closed were compared by a paired t-test. Fluxes were log-transformed which is a common prerequisite for analysing denitrification data due to its skewed distribution (Folorunso and Rolston, 1984). The measured additional $N_2+N_2O$ flux was compared with the modelled value with a one-sample t-test. Multiple regression analysis was conducted to derive a model of $N_2+N_2O$ production. T-tests and regression analysis were conducted with WinSTAT and SAS, respectively.

.

### 3 Results

### 3.1 Modelling

### 3.1.1 Surface and subsurface fluxes before and after chamber closure

Modelling results of Scenario *A_bottom_open* (imitating the field set-up) demonstrated 3D spatial distribution of gas concentrations and the resulting diffusive fluxes with highest concentrations in the centre of the [15]N-labelled soil volume with open chamber at steady state (Fig. 2a, Fig. S1). Time until steady state after the onset of [15]($N_2+N_2O$) production increased with decreasing gas diffusivity and increasing soil moisture. For soil water content (SWC) of 0.34 g g$^{-1}$, it was approximately 3 hours (Fig. 2a). Soil air concentration of [15]($N_2+N_2O$) at steady state also increased with SWC (data not shown).

Chamber closing leads to an increase of maximum concentrations (Fig. S1) and also to lowering of surface fluxes (Figs. 3 and 4).

After chamber closing, surface flux decreases continuously while subsurface flux increases and the storage flux initially increases before gradually decreasing. This shows that the lowering of surface flux with increasing time of chamber closing results from increasing subsoil flux but also from further accumulation of denitrification products in pore space. While surface flux is largest among all fluxes at chamber closing, it is exceeded by subsoil flux after about one hour. With increasing SWC, and thus decreasing diffusivity, the change in fluxes with time is lowering (Fig. 4). Highest relative subsoil fluxes are thus obtained at lowest SWC. For $N_2O$, the decrease in surface flux is slightly lower compared to $N_2$ (Fig. 4). The change in relative fluxes is almost identical for the different isotopologues of $N_2$ and $N_2O$ (only shown for $N_2$ in Fig. S3).

To understand the effect of the labelling design, modelled fluxes of scenario *B_30_30, B_45_30, B_45_45, B_60_45* were compared. With decreasing depth of [15]N-labelling, surface flux during the first hours after chamber closing increases, since less denitrification products accumulate or are lost to the subsoil. This is evident by comparing fluxes obtained with 30 cm and 45 cm depth of confined [15]N-labelled soil (Fig. 5). Increasing cylinder length below the depth of labelled soil, e.g., if the length of the cylinder extends 15 cm below the 30 cm or 45 cm deep labelled soil, yields an increase in surface flux and slight decrease in subsoil flux due to more accumulation of [15]N-labelled gases below the [15]N-labelled soil. Hence, underestimation of production based on surface flux is more severe with deeper labelling, but is lowering if the depth of confinement is increased. Modelled underestimation of $N_2$ production derived from chamber accumulation is summarized in Table 3. Depending on diffusivity, depth of [15]N-labeling and depth of confinement, underestimation ranges between 28 and 71 %. Possible deviations of these estimates that would result from errors in the determination of diffusivity can be seen by comparing the modelled underestimation at different SWC, giving a range of, e.g., 51 to 61% for the *B_30_30* scenario with 2 h closure.

**Table 3. Underestimation of N$_2$ production by chamber measurements using linear regression over time and mean subsoil loss of N$_2$ produced within the $^{15}$N-labelled soil. Underestimation and subsoil loss are relative to production rates of the labelled core.**

| | Scenario | *B_30_30* | | *B_45_30* | | *B_45_45* | | *B_60_45* | |
|---|---|---|---|---|---|---|---|---|---|
| Soil Water Content | Closure time | Underest. Chamber | Subsoil loss | Underest. Chamber | Subsoil loss | Underest. Chamber | Subsoil loss | Underest. Chamber | Subsoil loss |
| 0.24 | 1h | 57% | 36% | 45% | 21% | 55% | 42% | 53% | 28% |
| 0.24 | 2h | 61% | 38% | 49% | 22% | 59% | 44% | 56% | 28% |
| 0.24 | 6h | 71% | 47% | 59% | 27% | 67% | 51% | 65% | 32% |
| 0.34 | 1h | 53% | 36% | 41% | 22% | 52% | 44% | 50% | 29% |
| 0.34 | 2h | 55% | 37% | 43% | 22% | 55% | 45% | 52% | 29% |
| 0.34 | 6h | 61% | 42% | 50% | 25% | 60% | 48% | 57% | 31% |
| 0.44 | 1h | 51% | 40% | 42% | 26% | 55% | 51% | 53% | 34% |
| 0.44 | 2h | 51% | 40% | 42% | 26% | 56% | 51% | 53% | 34% |
| 0.44 | 6h | 53% | 41% | 44% | 26% | 57% | 51% | 55% | 35% |

If diffusion to the subsoil was omitted, e.g. by closing the bottom of cylinders in the field, or during laboratory incubations, soil air concentrations and surface fluxes increase (Fig. 6). When comparing values with and without omitted subsoil diffusion, relative surface flux two hours after closure was 0.75 and 0.4, respectively. But with bottom closed, surface flux was still significantly lower than production due to continuing pore space accumulation (relative storage flux of 0.25 after 2 hours).

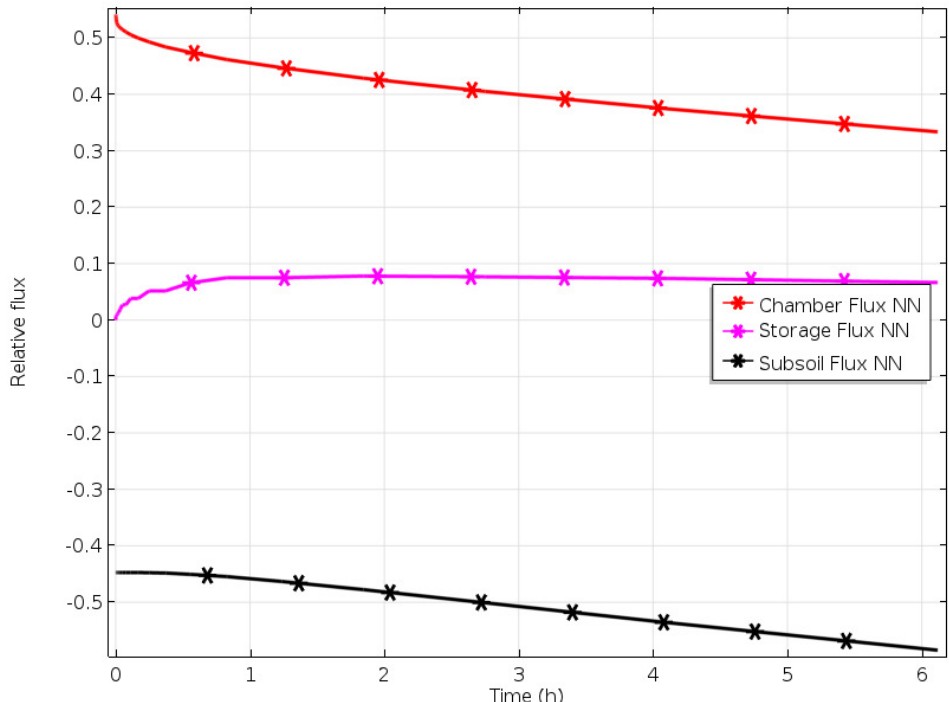

**Figure 3:** **Relative fluxes of $^{15}N^{15}N$ after chamber closing in scenario *A_bottom_open*.**

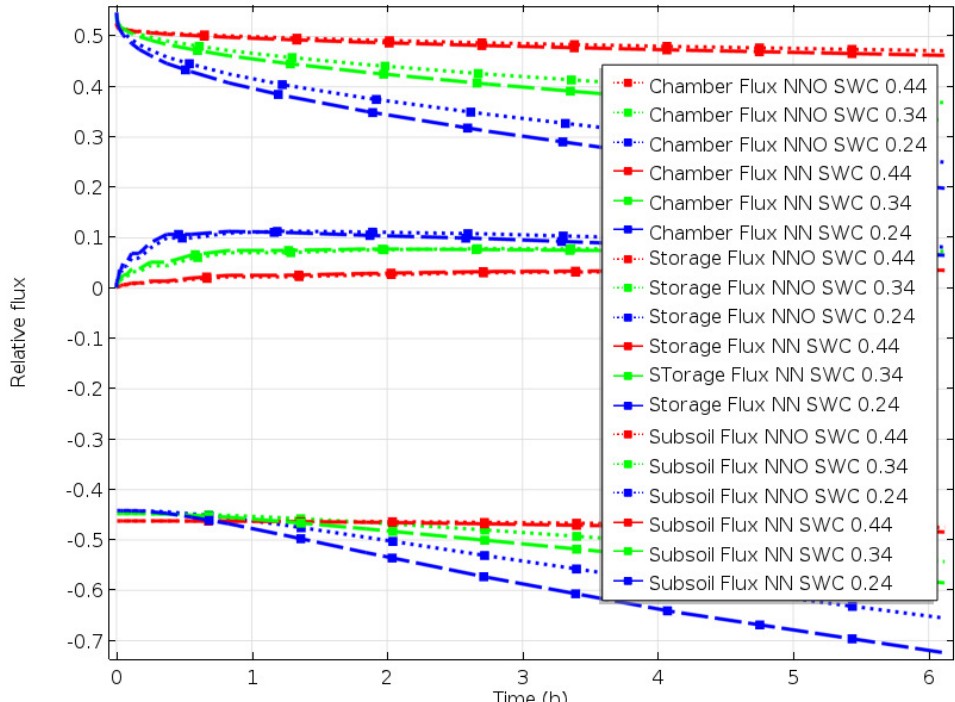

**Figure 4: Relative fluxes following chamber closing with different water contents in scenario *A_bottom_open* (surface flux, storage flux and subsoil flux starting positive, at zero and negative, respectively).**

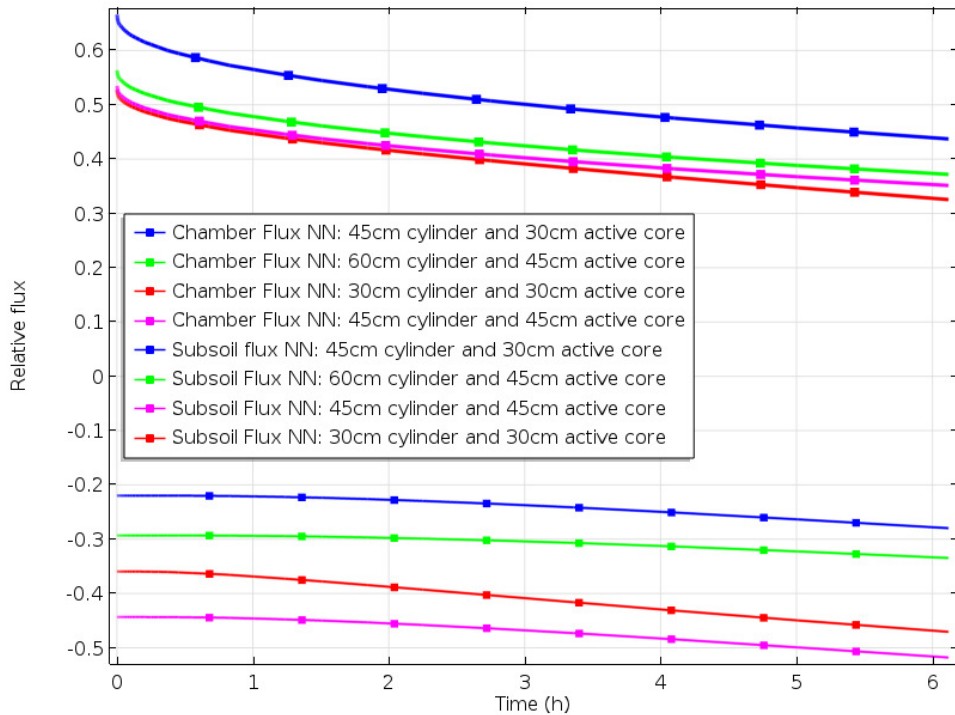

**Figure 5: Impact of the depth of the active core (representing depth of $^{15}$N labelling) and/or length of cylinder on relative surface and subsurface fluxes (scenarios *B_30_30, B_45_30, B_45_45, B_60_45* )**

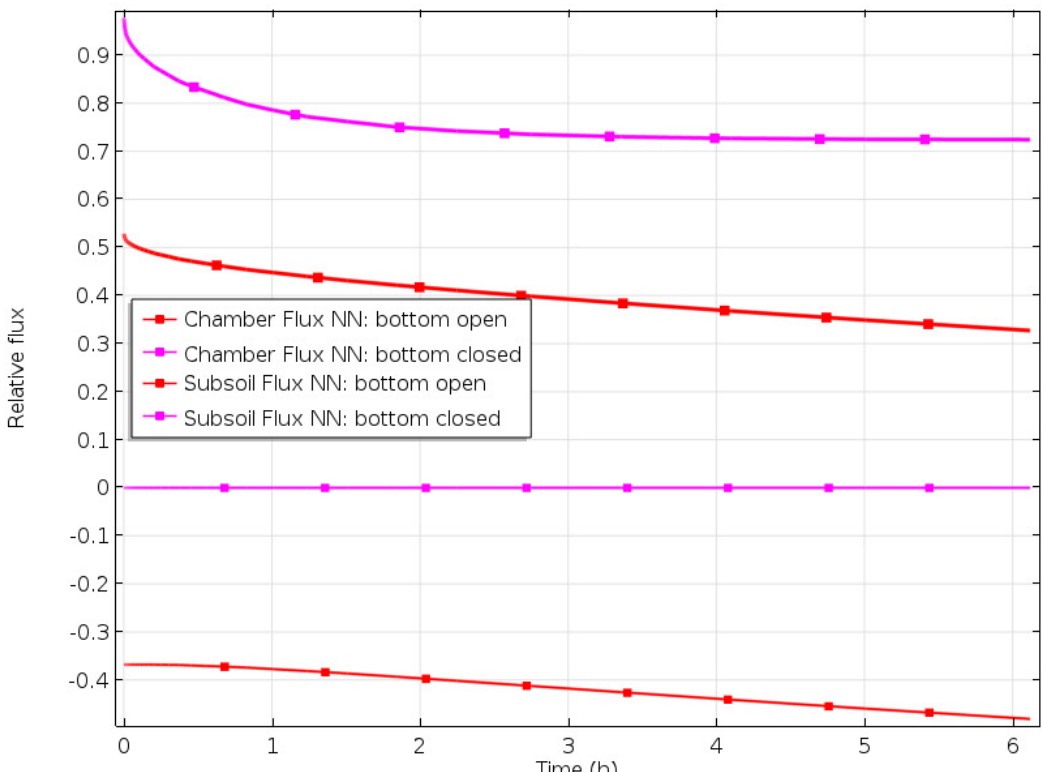

**Fig. 6: Simulated time course of surface and subsurface fluxes with bottom open or closed bottom (scenario *A_bottom_open* and *A_bottom_closed* at SWC = 0.35).**

**3.1.2 Determination of denitrification rates based on chamber concentration**

We conducted several runs using the field scenarios *A_bottom_open* and *A_bottom_closed* to generate a dataset that allowed the parametrization of functions that describe the relation of the concentration reached after 2h of chamber closure and the production within the labelled soil volume. This was done to allow a comparison of modelled data with the field measurements

(Table 5). Moreover, we hereby give an example how denitrification rates can be calculated using empirical equations, and thus without the need to run the 3D-model for each data evaluation.

We obtained the following equation to calculate the production of each gas species of interest (x = $^{14}N^{14}N$, $^{14}N^{15}N$, $^{15}N^{15}N$, $^{14}N^{14}NO$, $^{14}N^{15}NO$ or $^{15}N^{15}NO$) based on chamber concentration after a certain time of closure:

$$P_x = C_x \times \alpha + \delta \times C_x \times D_s \tag{1}$$

where $P_x$ is the production for the respective gas species (L ha$^{-1}$ d$^{-1}$), $C_x$ is concentration in ppm, $D_s$ the apparent gas diffusion coefficient (cm$^2$ s$^{-1}$) and α and δ are fit parameter.

**Table 4: Coefficients for the calculation of denitrification rates using Eq. 1 based on chamber concentrations for two hour chamber closing time, 30 cm depth of $^{15}N$ labelling, chamber height of 15 cm and 15 cm diameter and assuming subsoil diffusivity identical to diffusivity of the $^{15}N$-labelled soil. Coefficients α and δ were derived by regression analysis of modelled concentration ($R^2 > 0.999$).**

| | α | δ |
|---|---|---|
| **Production, Bottom open** | 35.4975± 0.012 | 432.9 ± 0.797 |
| **Production, Bottom closed** | 18.9469± 0.0133 | 219.4 ± 0.9026 |
| **Subsoil flux, bottom open** | -15.282± 0.0131 | -220.2±0.5983 |
| **Surface flux, bottom open** | 16.8918 ± 0.00034 | 1.3695 ±0.8974 |
| **Surface flux, bottom closed** | 16.8918 ± 0.00262 | 1.3655 ±0.0178 |

**3.2 Field measurement**

**3.2.1 Soil moisture, mineral N and bulk density**

Average $NO_3^-$-N was 16 mg N kg$^{-1}$ without significant trends with depth and no significant differences between cylinders (Table S1). $NH_4^+$-N was highest in 0 to 10 cm depth (1.8 mg N kg$^{-1}$) and < 1 mg N kg$^{-1}$ below 10 cm depth. Average $^{15}N$ atom fraction of extracted $NO_3^-$ ($^{15}a$) was 0.15, but values increased with depth in all cylinders, where 20-30 cm averages (0.2) were more than twice compared to 0-10 cm depth (0.09). Bulk density ranged between 1.48 and 1.52 with highest values at 10 to 20 cm depth. WFPS was higher at 0 to 10 cm depth (72%) than at 10 to 30 cm depth (60 to 62 %) with similar depth trends in all cylinders.

**Table 5: N₂+N₂O fluxes of field experiments with 2 hours chamber closing with and without closed bottom in comparison with modelled data.**

| # | Type and determination of data | Result |
|---|---|---|
| 1 | Measured $N_2 + N_2O$ surface flux, bottom open (g N ha$^{-1}$ d$^{-1}$) | 589 ± 284 |
| 2 | Measured $N_2 + N_2O$ surface flux, bottom closed (g N ha$^{-1}$ d$^{-1}$) | 805 ± 369 |
| 3 | % additional measured surface flux with bottom closed [100×(#2-#1)/#2] | 36.7 |
| 4 | Modelled relative surface flux, bottom open | 0.469 |
| 5 | Modelled relative surface flux, bottom closed | 0.879 |
| 6 | % additional modelled surface flux with bottom closed [100×(#5-#4)/#4] | 88.2 |
| 7 | Modelled $N_2 + N_2O$ production (g N ha$^{-1}$ d$^{-1}$) | 1055 |
| 8 | Modelled relative subsoil flux, bottom open | 0.432 |
| 9 | Modelled relative storage flux, bottom open | 0.099 |
| 10 | modelled relative storage flux, bottom closed | 0.121 |

5   **3.2.2 Field fluxes**

The comparison between surface flux with or without closing the cylinder bottom was conducted on June 4, 2016 with chamber closing at 10:40 AM (bottom open) and 2:40 PM (bottom closed). Mean surface flux of $N_2+N_2O$ with bottom open was 589 g N ha$^{-1}$ d$^{-1}$ (Table 4) and thus in between the fluxes observed during preceding two days (460 ± 161 g N ha$^{-1}$ d$^{-1}$ on June 2, 6:50 PM; 657 ± 206 g N ha$^{-1}$ d$^{-1}$ on June 3 at 11 AM). This shows that denitrification rates were quite stable over several days and

10   that denitrification was a significant N loss, probably due to the coincidence of high soil moisture and NO₃⁻ content (Table S1). The residual fraction of N₂O remaining after N₂O reduction to N₂ ($r_{N2O}$) was 0.15 on average (Table S2), showing that N₂ dominated $N_2+N_2O$ fluxes. Mean ¹⁵a values for each cylinder were somewhat variable (0.09 to 0.18). Means of ¹⁵a (Table S1) and of the ¹⁵N enrichment of the labelled N pool producing N₂O (a_{p_N2O}, Table S2) were in close agreement (0.15 and 0.16, respectively).

Comparing $N_2+N_2O$ surface fluxes when the cylinders were open or closed at the bottom resulted in significantly ($P < 0.05$) higher surface fluxes when closed (Table 5) which was evident for each of the replicate micro-plots (Table S2). Because bottom-closed measurement was conducted as soon as possible immediately after the bottom-open measurement, i.e. after venting of the cylinders with chambers open for two hours and thus four hours after bottom open measurements, we assumed that denitrification rates had not changed significantly and the increase in surface fluxes was due to bottom closing.

The $a_{p\_N2O}$ values of bottom-open and bottom-closed measurements exactly coincided. Conversely, the $N_2O$ residual fraction ($r_{N2O}$) of individual cylinders differed inconsistently since $r_{N2O}$ of bottom-closed measurements were higher in replicates 1 and 4, but were lower in replicates 2 and 3.

## 3.3 Comparison of modelled and measured surface flux

The ability of the model to predict the time pattern of gas accumulation was evaluated by comparing measured and simulated values. Model runs using the Ds values calculated from measured moisture and bulk density data of the field experiment assuming open or closed bottom yielded relative surface fluxes of 0.47 and 0.88, respectively (Table 5). The additional surface flux with bottom closed was thus quite relevant according to both, model and measurement. However, the magnitude of the modelled additional flux (88%) was more than twice, and thus significantly higher ($P < 0.001$) compared to the measured value. Using Eq. 1 and $N_2+N_2O$ concentration in the chamber measured in the field with open cylinder bottom and using respective coefficients of Table 1 resulted in $N_2+N_2O$ production of 1055 g N $ha^{-1}$ $d^{-1}$. The modelled subsurface flux with bottom open was almost half of the $N_2+N_2O$ production. Modelled accumulation of $N_2+N_2O$ in the pore space of the $^{15}N$-labelled soil was higher with bottom closed (relative storage flux of 0.12) compared to bottom open (relative storage flux of 0.10). Evaluation of $N_2$ and $N_2O$ fluxes individually yielded results similar to $N_2+N_2O$ fluxes (data not shown).

## 4 Discussion

### 4.1 Field study

Our comparison between $^{15}(N_2+N_2O)$ fluxes from $^{15}N$-labelled micro-plots with and without closing the bottom of the cylinders supplied for the first time direct evidence for the underestimation of $^{15}(N_2+N_2O)$ production due to diffusive loss to the subsoil as suggested earlier (Mahmood et al., 1998; Sgouridis et al., 2016). In view of the poor sensitivity of the $^{15}N$ gas flux method in the field under ambient atmosphere (Well et al., 2018), a prerequisite for this proof was the occurrence of sufficiently high and relatively stable denitrification rates. These conditions were given in our experiment due to the coincidence of high soil moisture and $NO_3^-$-N during the experimental period. Considering the relatively low variation of denitrification rates during

two days preceding the comparison, we conclude that the increase in surface fluxes after closing of the cylinder bottom was mainly due to the exclusion of diffusive loss to the subsoil.

While the increase in $^{15}(N_2+N_2O)$ surface flux after bottom closing was comparable among the four replicates, this was not the case for the $N_2O$ flux and $r_{N2O}$, which both exhibited considerable variabilities. $r_{N2O}$ showed larger deviation probably because the $N_2O$ reduction to $N_2$ is not only sensitive to $N_2O$ concentration in pore space, but also to changes in control factors like temperature, $O_2$, $NO_3^-$ and labile C (Mueller and Clough, 2014). We suspect that the latter factors were somewhat variable within the replicates and that their interaction with $N_2O$ concentration lead to the observed variability in $r_{N2O}$. The apparent sensitivity of $r_{N2O}$ to bottom closing shows that care should be taken when interpreting $N_2O$ reduction to $N_2$ from $r_{N2O}$ determined in closed laboratory systems. Apart from our observations, an effect of bottom closure on $N_2O$ reduction is to be expected since the resulting increased pore space $N_2O$ concentration would favour $N_2O$ reduction. This effect would thus lead to overestimation of $N_2O$ reduction when extrapolating results to the field.

## 4.2 Estimating production of N₂ and N₂O based on surface fluxes and diffusion modelling

Modelling diffusive fluxes of $N_2$ and $N_2O$ evolved from $^{15}N$-labelled soil showed that denitrification rates are underestimated by more than 50% when only surface fluxes are taken into account, which has been general practice in the past (Sgouridis et al., 2016 and references therein). Modelling also confirmed that in contrast to our hypothesis, not only subsoil flux is a relevant fraction of $^{15}(N_2+N_2O)$ production, but also the increasing accumulation during chamber closing. Several authors increased the chamber deployment time of 40 to 60 minutes as common for $N_2O$ flux measurement (Parkin et al., 2012), e.g., to 2 hours (Tauchnitz et al., 2015; Buchen et al., 2016) or even 24 hours (Sgouridis et al., 2016). This was done to increase $^{15}N_2+N_2O$ concentration in the chamber and thus to improve the detection limit for denitrification at a given IRMS precision. Because surface fluxes are lowering with deployment time, it is clear that the underestimation of surface flux based denitrification rates is also increasing.

For laboratory studies with the $^{15}N$ gas flux method using closed incubation systems, our findings on $^{15}N_2$ and $^{15}N_2O$ accumulation in pore space is quite relevant. Closing incubation vessels for a limited time and estimating denitrification from headspace concentration (e.g Meyer et al., 2010; Siegel et al., 1982) inevitably leads to underestimation of denitrification rates. Experimental evidence for this underestimation was obtained by destroying pore structure at final sampling to homogenize headspace and pore space (Harter et al., 2016). Because the fraction of denitrification products accumulated in pore space increases with decreasing diffusivity, the problem is most severe for water-saturated soils. While this had also previously been solved by homogenizing headspace and pore space before sampling (Well and Myrold, 1999), other studies with water-saturated substrates did not take accumulated gases into account (e.g. Nielsen, 1992).

Our results show that extending chamber deployment time is not a good strategy to improve the detection limit for denitrification. This is because the fraction of gaseous denitrification products that is not emitted at the soil surface is increasing with time. Although we can now estimate this fraction with our model, uncertainties of the modelled data lead to increasing uncertainty in denitrification estimates with chamber deployment time. Another way to improve detection is to lower the $N_2$

background concentration in the field by flushing chambers with an $N_2$-depleted gas matrix (Well et al., 2018). Due to the good sensitivity of that method, chamber deployment could be kept at one hour. Principally, our modelling approach could also determine the subsurface flux and pore space accumulation for that method, but will have to be adapted to take diffusion dynamics in the $N_2$-depleted gas matrix into account.

Because the flux dynamics of gaseous denitrification products in the soil had not been taken into account in past field flux and certain laboratory studies, we assume that numerous studies underestimated denitrification significantly. It can thus be concluded that soil denitrification is probably even more relevant than assumed today.

Our model approach is suitable to estimate pore space accumulation and subsoil diffusion of denitrification products. It thus

allows to determine production based on surface fluxes in field flux studies but also in closed laboratory incubations. Principally, it could also be used to correct previously published data if necessary information on diffusivity and pore space was available. For experiments with the same dimensions and bulk density as assumed in our regression model it is also possible to calculate production from surface flux using the parameters of Table 4. Principally, the regression approach offers an easy way to derive production without the need to run the 3D model. But to obtain a general solution that would fit any

experimental conditions in terms of bulk density, depth of labelling, chamber design and deployment time, it will be necessary to conduct multiple model runs, which was beyond the scope of this paper.

Our approach includes several factors of uncertainty. Prerequisite for precise quantification is the knowledge of the vertical distribution in activity and diffusivity. Moreover, we have to assume steady state, which is never perfectly realized due to

temporal change of diffusivity and denitrification rates, e.g., following precipitation and thus decreasing diffusivity, increasing moisture and change in the labelled volume. Finally, we did not yet take into account water phase transport. But this has some relevance due to low diffusivity in the water phase. The impact of water phase transport should be largest for $N_2O$ due to its high solubility in water, yet gas diffusivity of $N_2O$ in water  is more than 3 orders of magnitude lower than in air (Rabot et al 2015). For $CO_2$, which has also a high solubility in water, the contribution of the aqueous phase to diffusive fluxes is negligible

when the ratio of air-filled porosity and total pore-space is greater than 0.12 (Jassal et al., 2004). But since denitrification occurs often in soil near water saturation, water phase dynamics might be another explanation for the deviations between the $N_2O/(N_2+N_2O)$ ratios determined with bottom open and bottom closed.

The general agreement between measured and modelled increase in surface flux after closing the cylinder bottom can be seen as a first proof of our concept to quantify denitrification rates using surface fluxes and modelling. Reasons for the observed deviations between experimental and model results can be manifold. In view of the aforementioned factors of uncertainty, these could include imperfect estimation of Ds by the empirical model (Millington, 1959), spatial variability of diffusivity (Kuhne et al., 2012; Lange et al., 2009; Maier et al., 2017; Maier and Schack-Kirchner, 2014; Marrero, 1972) within the 10 cm layers for which Ds was determined, spatial variability of denitrification rates (Groffman et al., 2009) and incompete steady state. Further reasons could be the production of $^{15}N_2$ and $^{15}N_2O$ from possibly leached $^{15}NO_3^-$ below the confined soil cores, and a possible shift in denitrification rates during the 6 hours between the two experiments with bottom open and bottom closed. A quantitative evaluation of the model by $^{15}N$ gas flux experiments would be quite challenging since it would mean to assess all aforementioned uncertain factors and to include heterogeneity in the modelling. Future attempts are therefore necessary to improve model evaluation and check how our approach will perform under heterogenic conditions. But despite these uncertainties, the general agreement of model and measurements shows that our approach leads to improved denitrification estimates.

Which progress in flux estimation is obtained in view of incomplete knowledge on parameters and could incorrect parameter estimation lead to augmented bias? Even uncertain estimates of subsoil fluxes would improve the outcome of the $^{15}N$ gas flux method in comparison with current practice (i.e. without taking subsoil diffusion and storage into account) as it would lower the bias in estimating denitrification rates. We can exclude that our approach would increase total bias in estimating denitrification through incorrect determination of diffusivity. A larger overestimation of subsoil diffusion or storage could only occur at high soil gas diffusivity, that means in dry highly porous soils (Table 3). But these conditions are less relevant for our approach since denitrification is inhibited at high diffusivity. Taking into account the uncertainty in subsoil diffusion modelling we demonstrate that worst case scenarios would still improve estimates compared to previous practice: the scenario with 30 cm depth of labelling and confinement of labelled soil (*B_30_30*) yields 51 and 61 % underestimation for highest and lowest modelled water content, respectively. Our approach would thus overestimate production by up to 10 % whereas production derived from surface flux only would underestimate the true production at least by 51 %. Between soil water contents of 0.34 and 0.44, our overestimation would be only 2 % (i.e. 53% - 51%). Consequently, potential bias of our correction approach arising from errors in determination of diffusivity would be quite small under conditions favouring denitrification. Under drier conditions, errors would still be much smaller compared to the errors from neglecting subsurface fluxes. The moderate impact of diffusivity also shows that spatial heterogeneity of diffusivity (Kühne et al. 2012) would not have a large impact and its assessment would not have to be prioritized.

While it was beyond the scope of this study to evaluate uncertainty in detail, future work should follow this up in order to explore the achievable accuracy in estimating subsoil flux and storage under given conditions. This should include modelling

water phase transport, depth of labelling and the impact of spatial and temporal variability in diffusivity and denitrification rates. Moreover, controlled experiments would be needed to validate model results as far as possible.

**5 Conclusions**

Measurements and production-diffusion modelling showed that field surface fluxes of [15]N-labelled $N_2$ and $N_2O$ emitted from
[15]N-labelled soil $NO_3^-$ severely underestimate denitrification due to subsoil flux and accumulation in pore space. The extent of underestimation increases with chamber deployment time. Soil denitrification has thus been underestimated in many previous studies using the [15]N gas flux method without taking subsoil flux and accumulation in pore space into account. While production-diffusion modelling is a promising tool to estimate subsoil flux and storage flux, the observed deviations between experimental and modelled subsoil flux reveal the need for refined model evaluation. To enable correction of previously
published data, further model parametrization work should cover all soil and land use types.

**Author contribution**

RW designed the overall concept. RW and DL designed the field experiments and DL and NR carried them out. MM developed the model code and performed the simulations. RW prepared the manuscript with contributions from JK, DL and MM.

**Acknowledgements**:

This study was funded by the Deutsche Forschungsgemeinschaft through the project LE 3367/1-1 and the research unit 2337: "Denitrification in Agricultural Soils: Integrated Control and Modeling at Various Scales (DASIM)". We thank Frank Hegewald for technical support in experiments, Martina Heuer and Jennifer Ehe for stable isotope analysis, Kerstin Gilke and
Andrea Oehns-Rittgerod for analysis by GC, and Roland Fuß for support in statistical analyses. We further thank for supply of an experimental field site by the Faculty of Agriculture, University of Applied Sciences Southern Westphalia.

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
