# Peer review of "Underestimation of denitrification rates from field application of the $^{15}\text{N}$ gas flux method and its correction by gas diffusion modelling"

_Biogeosciences, 2018_

## Referee Comment (RC1) · Anonymous Referee #1 · 20 Dec 2018

Review of Biogeosciences [Manuscript #2018-495] Title: Underestimation of denitrification rates from field application of the 15N gas flux method and its correction by gas diffusion modelling.

Dear Associate Editor, The manuscript in consideration sought to elucidate whether the field application of the 15N gas flux method underestimates denitrification rates and evaluate the possible reasons by using soil diffusion modelling. The topic is of particular interest to terrestrial biogeochemists attempting to constrain N cycling processes since denitrification is notoriously difficult to measure under field conditions. The authors significantly advance knowledge in this field of research by providing a

[Figure]

Interactive
comment

first proof of denitrification rates underestimation due to subsoil diffusion and storage of denitrification products as stipulated previously by research published in this journal. Even though the authors provide strong indications that subsoil diffusion is indeed occurring during field application of the 15N gas flux method I am not convinced of the practical applicability of soil diffusion modelling for correcting this discrepancy. The significant difference between measured and modelled results suggests there is too many unknown factors (e.g. spatial variability of diffusivity) and that further assumptions (beyond the homogenous soil labelling of the 15N gas flux method) need to be introduced to model surface flux and subsoil diffusion and storage. I am wondering if the soil diffusivity assumptions (homogenous soil pore structure and water content, absence of stones, roots etc and constancy of diffusion and production rates) are actually introducing more bias than practical improvements to the traditional chamber method (e.g. depth of labelling, size of chamber, open or closed bottom and closure time). The authors suggest that previously published data could be corrected for underestimation by using their model with further parameterisation. This indeed would be something I would be very interested to see and particularly for more challenging soil types than arable land such as grasslands or forests. The manuscript is well structured and clearly written and it seems to me it is a first step towards the right direction for further improving field denitrification measurements. I therefore recommend that the manuscript is accepted for publication following a few minor corrections and clarifications detailed below: Minor comments: 1. P1 Lines 18: End the sentence after total production and start a new one after it. 2. P3 L9 and throughout: Please correct spelling of the word labelled throughout the manuscript. 3. P3 L25&26: In Sgouridis et al. 2016 the labelled nitrate was applied via injections to the soil volume. Please correct the reference and replace with one that surface application was used. 4. P4 L16: Was steady state within the first 6 hours after the label application also measured or just modelled? In the next sentence the assumption stated is that gas production starts at constant rates after the label application. Is it therefore necessary to first establish steady state before applying the model? 5. P10 L22: Reference is repeated twice. 6. P11 L26: It would have been

useful if measurements with or without closed bottom cylinder and varying labelling depths and lengths of cylinders were also taken. This could have shown whether the model predictions are true and if there is a significant difference in surface fluxes to justify the use of the model. Perhaps a combination of lower labelling depth, deeper cylinder and larger chamber would result in insignificant subsoil diffusion losses. 7. P19 L2: I agree that it would be a lot easier to apply the model under laboratory closed system conditions. However, pore space/headspace equilibration is relatively easier to achieve than attempting the soil diffusion modelling. The real challenge for the future application of the model would be to apply it under field conditions in more challenging soil types.
* * *

---

## Referee Comment (RC2) · Anonymous Referee #2 · 30 Jan 2019

Overall the paper is quite relevant to researchers who have used or are planning to use the 15N labelling method to quantify identification rates in-situ. The researchers convincingly show through modelling and field data that the impact of subsoil diffusion and storage fluxes have a significant impact on the estimated denitrification rates and thus have likely caused under reporting in the current literature.

General comments

- My main reservation is regarding the applicability of the modelling more broadly for correction of field results. Parameters like diffusivity are notoriously difficult to estimate in the field, and therefore the discrepancy between model and measured as reported here may never be reconcilable.

[Figure]

- While i appreciate the difficulty of including water phase gas transport in a model, especially one with such a complicated isotopologue structure, I feel it should at least be discussed in the paper as another important factor. It would both contribute to pore space storage as well as isotopic fractionation although the latter may not be important given the label strength.

- The model results are somewhat dense and difficult to digest - my concern is that someone who is not a modeller/gas diffusion specialist would get lost in the current brief narrative. Suggest being more verbose but for the benefit on enhanced clarity.

- Section 3.1.2 requires significantly more explanation. I would have expected to see a more normal flux calculation as a proxy for production as is done with CO2 or CH4, however the fitting approach is applied here. Why did the authors not use a linear or exponential flux model as is commonly used for other gases. What do the parameters alpha and delta signify or what is their physical manifestation - are they related to chamber volume and surface area, cylinder depth, etc? Is this approach/equation commonly applied outside of this paper?

- Discussion and conclusions - If the modelling approach cannot be applied quite yet to correct the values, perhaps there should be a small table or histogram or similar of "likely errors" that may have been incurred in past experiments using this method. This would at least allow the community to make an educated guess on how far off our current estimates are from reality (and may allow some reconciliation across methods as well).

- Overall the flow of the paper could be improved, this is partly due to sections with poor sentence structure or run-on thoughts mostly in the introduction and discussion portion of the paper.

Specific Comments Page 2
- Line 9 - "to measure" should be "in measuring" or similar. - Line 14 - gastight should be gas tight unless this is a brand of container - Line 25 - suggest inserting several sentences explaining to the reader why in-situ measurements are important. Is there

literature to cite comparing in-situ to lab incubations or similar? Page 3
- Line 8 - some more detail around why we don't just measure these parameters instead of modelling them Page 4
- Line 14 - change amount to concentration - Lines 18-29 - consider separating into bullets. -Consider annotating figures with some of the details contained in lines 18-29 Page 6
- Figure 2b - What is the origin of the oscillation in the flux data Page 7
Line 1- is the chamber here fully/homogeneously mixed? Line 18 - Is the atmosphere multi-layer? This isn't clear Line 22 (and elsewhere) - NO3 is often used, but are there any chemical or biological processes modelled that convert NO3 to other species? If not then perhaps its best to clarify that gases are produced independent of NO3 transformation. Page 8
Line 10 - Is production constant with depth over the length of the collar? Page 11
- Line 11 - Clarify that these initial results are from the bottom open scenario

---

## Author Comment (AC1) · 21 Feb 2019

We thank the reviewers for their constructive comments (repeated under RC1.x), which revealed several gaps and options for improvement. They will be the basis for substantial improvement of the manuscript. In the following we answer all comments and declare how we will change the manuscript to fulfill the comments.
cation rates from field application of the 15N gas flux method and its correction by gas diffusion modelling. Dear Associate Editor, The manuscript in consideration sought to elucidate whether the field application of the 15N gas flux method underestimates denitrification rates and evaluate the possible reasons by using soil diffusion modelling. The topic is of particular interest to terrestrial biogeochemists attempting to constrain N cycling pro- cesses since denitrification is notoriously difficult to measure under field conditions. The authors significantly advance knowledge in this field of research by providing a first proof of denitrification rates underestimation due to subsoil diffusion and storage of denitrification products as stipulated previously by research published in this jour- nal. RC1.1 Even though the authors provide strong indications that sub-soil diffusion is indeed occurring during field application of the 15N gas flux method I am not convinced of the practical applicability of soil diffusion modelling for correcting this discrepancy. The significant difference between measured and modelled results suggests there is too many unknown factors (e.g. spatial variability of diffusivity) and that further assump tions (beyond the homogenous soil labelling of the 15N gas flux method) need to be introduced to model surface flux and subsoil diffusion and storage. I am wondering if the soil diffusivity assumptions (homogenous soil pore structure and water content, absence of stones, roots etc and constancy of diffusion and production rates) are actually introducing more bias than practical improvements to the traditional chamber method (e.g. depth of labelling, size of chamber, open or closed bottom and closure time).

AC1.1 Response: We agree that modelling of subsoil fluxes is associated with a variety of uncertainties which we explained in P 19 L 20-31: "The general agreement between measured and modelled increase in surface flux after closing the cylinder bottom is a first proof of our concept to quantify denitrification rates using surface fluxes and mod-elling. Reasons for the observed deviations between experimental and model results can be manifold, e.g., imperfect estimate of Ds by the empirical model (Millington & Quirk), spatial variability of diffusivity (Kuhne et al., 2012; Lange et al., 2009; Maier et al., 2017; Maier and Schack-Kirchner, 2014; Marrero, 1972) within the 10 cm layers

for which Ds was determined, spatial variability of denitrification rates (Groffman et al., 2009), production of 15N2 and 15N2O from possibly leached 15NO3- below the confined soil cores, and a possible shift in denitrification rates during the 6 hours between the two experiments with bottom open and bottom closed. A quantitative evaluation of the model by 15N gas flux experiments would be quite challenging since it would mean to assess all aforementioned uncertain factors and to include heterogeneity in the modelling. Future attempts are therefore necessary to improve model evaluation and check how our approach will perform under heterogenic conditions. But despite these uncertainties, the general agreement of model and measurements shows that our approach leads to improved denitrification estimates. " But even uncertain estimates of subsoil fluxes would improve the outcome of the 15NGF in comparison with current practice (i.e. without taking subsoil diffusion and storage into account) as it would lower the bias in estimating denitrification rates. We can exclude that our approach could increase bias in view of the limited effect by varying diffusivity on relative subsoil diffusion and storage flux (see also AC 2.6 below). Severe overestimation of these quantities could only occur at high soil gas diffusivity, that means in dry highly porous soils . But these conditions are not relevant for our approach since denitrification is inhibited at high diffusivity.

Changes: We will address the points above in the extended discussion.

RC1.2 The authors suggest that previously published data could be corrected for underestimation by using their model with further parameterisation. This indeed would be something I would be very interested to see and particularly for more challenging soil types than arable land such as grasslands or forests. AC1.2 Response: Thank you for this suggestion. We plan to do this in follow-up studies Changes: In the conclusions we will mention that follow-up studies are needed to obtain further model parametrization to enable correction of previous published in situ N2+N2O fluxes covering all land use types

The manuscript is well structured and clearly written and it seems to me it is a first step

towards the right direction for further improving field denitrification measurements. I therefore recommend that the manuscript is accepted for publication following a few minor corrections and clarifications detailed below: Minor comments: RC1.3 1. P1 Lines 18: End the sentence after total production and start a new one after it.

AC1.3 Response/Changes: will be done as suggested

RC1.4 2. P3 L9 and throughout: Please correct spelling of the word labelled throughout the manuscript. AC 1.4 Response/Changes: will be done as suggested

RC1.5 3. P3 L25&26: In Sgouridis et al. 2016 the labelled nitrate was applied via injections to the soil volume. Please correct the reference and replace with one that surface application was used. AC1.5 Response/Changes: will be done as suggested

RC1.6 4. P4 L16: Was steady state within the first 6 hours after the label application also measured or just modelled? In the next sentence the assumption stated is that gas production starts at constant rates after the label application. Is it therefore necessary to first establish steady state before applying the model? AC1.5 Response: Indeed the model can only yield correct values if fluxes are steady state at chamber closing. Hence, surface flux data collected immediately after labelling where activity and fluxes dramatically change over time could not be corrected exactly. Moreover, we did not determine steady state experimentally. However, measurements were conducted 5 days after labelling (section 2.3). Therefore, since this is 20 times the modelled steady state time, we can expect that near steady state was reached, even though we could not check this. But we realize that incomplete steady state, due to changes in activity and/or diffusivity, e.g., following precipitation, could be an issue. We will include this in the discussion. Changes: In the discussion we will add a statement that further uncertainty could arise from incomplete steady state , e.g., following precipitation and thus decreasing diffusivity, increasing moisture and change in the labelled volume, and that this effect should be evaluated in follow-up studies. RC1.6. P10 L22: Reference is repeated twice. AC1.6 Response/Changes: this will be corrected

RC1.7 P11 L26: It would have been useful if measurements with or without closed bottom cylinder and varying labelling depths and lengths of cylinders were also taken. This could have shown whether the model predictions are true and if there is a significant difference in surface fluxes to justify the use of the model. Perhaps a combination of lower labelling depth, deeper cylinder and larger chamber would result in insignificant subsoil diffusion losses. AC1.7 Response: This is indeed planned for a follow up study Changes: In the discussion we will explain that the present study only compared constant depth of labelling and that the applicability of our approach for varying depth of labelling should be checked by future studies.

RC1.8 P19 L2: I agree that it would be a lot easier to apply the model under laboratory closed system conditions. However, pore space/headspace equilibration is relatively easier to achieve than attempting the soil diffusion modelling. The real challenge for the future application of the model would be to apply it under field conditions in more challenging soil types.

AC1.8 Response: We agree. Closed system was mentioned because there is also some effect that had not been taken into account until now. The challenge for the future application of the model would be to apply it under field conditions in more challenging soil types as was already addressed in the following section in page 19, where we discuss the potential and limitation of our approach for field studies. Changes: In the discussion we will deepen the discussion on future application of the model to apply it under field conditions in more challenging soil types.

---

## Author Comment (AC2) · 21 Feb 2019

We are grateful to the Editors for considering our manuscript. We thank the reviewers for their constructive comments (repeated under RC2.x), which revealed several gaps and options for improvement. They will be the basis for substantial improvement of the manuscript. In the following we answer all comments and declare how we will change the manuscript to fulfill the comments.

Anonymous Referee #2 Overall the paper is quite relevant to researchers who have used or are planning to use the 15N labelling method to quantify identification rates in-situ. The researchers convincingly

show through modelling and field data that the impact of subsoil diffusion and storage fluxes have a significant impact on the estimated denitrification rates and thus have likely caused under reporting in the current literature.

General comments

R2.1 - My main reservation is regarding the applicability of the modelling more broadly for correction of field results. Parameters like diffusivity are notoriously difficult to estimate in the field, and therefore the discrepancy between model and measured as reported here may never be reconcilable.

AC2.1 Response: This was also addressed by reviewer 1 (RC1.1). We agree that exact prediction of subsoil diffusion is difficult, but determination of Ds in the subsoil is feasible with reasonable effort and can also be modelled based on bulk density and soil moisture. This will lead to Ds estimates accurate enough for subsoil diffusion modeling (see als AC2.6 below).

Changes: In the results we will add additional scenarios to illustrate the impact from uncertain estimates of diffusivity. In the discussion we will add a section to explain that (i) taking into account maximum uncertainty in subsoil diffusion modelling we demonstrate that worst case scenarios would still improve estimates compared to previous practice and (ii) that small scale heterogeneity in Ds had little or moderate effect on simulate subsoil diffusion. . ..". Moreover, we will supply references highlighting the accuracy of Ds measurement and modelling.

R2.2 - While i appreciate the difficulty of including water phase gas transport in a model, especially one with such a complicated isotopologue structure, I feel it should at least be discussed in the paper as another important factor. It would both contribute to pore space storage as well as isotopic fractionation although the latter may not be important given the label strength. AC2.2 Response: We agree, especially in case of N2O due its high solubility in water, there would be some storage which would have some impact on the change of fluxes following chamber closing due to the slow diffusion in the

water phase. This should be tested in follow up studies. Changes: In the discussion we will mention that water phase transport was not yet taken into account, but might have some relevance due low diffusivity in the water phase. We will also mention that this effect would be largest for N2O due to its high solubility in water and will add references were the impact of water phase gas transport is addressed with respect to CO2 flux modelling. We will also mention that water phase dynamics might be another explanation for the deviations between the N2O/(N2+N2O) ratios determined with bottom open and bottom closed.

RC2.3 - The model results are somewhat dense and difficult to digest - my concern is that someone who is not a modeller/gas diffusion specialist would get lost in the current brief narrative. Suggest being more verbose but for the benefit on enhanced clarity. AC2.3 Response / Changes: we are sorry for this. The model results will be explained more detailed.

RC2.4 - Section 3.1.2 requires significantly more explanation. I would have expected to see a more normal flux calculation as a proxy for production as is done with CO2 or CH4, however the fitting approach is applied here. Why did the authors not use a linear or exponential flux model as is commonly used for other gases. What do the parameters alpha and delta signify or what is their physical manifestation - are they related to chamber volume and surface area, cylinder depth, etc? Is this approach/equation commonly applied outside of this paper? AC 2.4 Response / Changes: we are sorry for this. We will supply adequate explanations.

- Discussion and conclusions

RC2.6 - If the modelling approach cannot be applied quite yet to correct the values, perhaps there should be a small table or histogram or similar of "likely errors" that may have been incurred in past experiments using this method. This would at least allow the community to make an educated guess on how far off our current estimates are from reality (and may allow some reconciliation across methods as well). AC 2.6 Response: We agree that this is would be useful. We will generate such a table based on the scenarios we already included in the manuscript, and add additional new scenarios Changes: We will add a table showing % underestimation for our micro-plot and chamber geometry in dependence of Ds and chamber deployment time. We will discuss how results would change with differing depth in labelling or size of the micro-plots.

RC2.7 - Overall the flow of the paper could be improved, this is partly due to sections with poor sentence structure or run-on thoughts mostly in the introduction and discussion portion of the paper. AC2.7 Response: We are sorry for this and will evaluate and improve the flow of the paper and check sentence structures. Changes: We will work on flow and sentence structure as requested

Specific Comments

Page 2 RC2.8 - Line 9 - "to measure" should be "in measuring" or similar.

AC2.8 Response/changes: will be done as suggested

RC2.9 - Line 14 - gastight should be gas tight unless this is a brand of container

AC2.9 Response/changes: will be done as suggested

RC2.10 - Line 25 - suggest inserting several sentences explaining to the reader why in-situ measurements are important. Is there literature to cite comparing in-situ to lab incubations or similar?

AC 2-10 Response: Denitrification is complexly controlled by interaction of labile C, abundance and community structure of denitrifiers, pore structure, soil and root respiration, mineral N dynmics. Hence, it is difficult to keep conditions in the lab identical to the field where some conditions dynamically change due to climatic factors but especially due to the activity of plants. Field measurements are therefore needed for reliable determination of denitrification in ecosystems. There are numerous lab studies but few field measurements. To the best of our knowledge respective comparisons are still missing for unsaturated soils. We conducted such a comparison only for denitrification in shallow groundwater (Well, R., et al. (2003). "Comparison of field and laboratory measurement of denitrification and N2O production in the saturated zone of hydromorphic soils." Soil Biology & Biochemistry 35(6): 783-799.

Changes: The content of the response will be included in the introduction.

Page 3 RC 2.11 - Line 8 - some more detail around why we don't just measure these parameters instead of modelling them AC 2.11 Response: we are not sure if we understand this question correctly. Line 8 f reads: "Modelling diffusion of 15N2 + 15N2O produced in 15N-lableled surface soil could be used to estimate its accumulation in pore space and diffusive loss to the subsoil and thus to quantify denitrification from the sum of surface flux, subsoil flux and storage within the 15N-labelled soil volume." By modelling diffusion we mean: modelling the diffusive flux. But the diffusive flux can not be measured directly. But unfortunately we failed to mention that this approach would include measurement of surface flux.

Changes: We will reformulate this sentence as: "Modelling diffusive fluxes of 15N2 + 15N2O produced in 15N-labeled surface soil based on measured surface flux and diffusivity could be used to estimate its accumulation in pore space and diffusive loss to the subsoil and thus to quantify denitrification from the sum of surface flux, subsoil flux and storage within the 15N-labelled soil volume. "

RC2.12 - Line 14 - change amount to concentration AC 2.12 Response/changes: will be done as suggested

RC2.13 - Lines 18-29 - consider separating into bullets. AC 2.13 Response/changes: will be done as suggested

RC2.14 -Consider annotating figures with some of the details contained in lines 18-29 AC 2.14 Response: This will be done as suggested: Changes: Modified caption of Fig. 2a,b: Figure 2a: Increase in pore space concentrations of N2 evolved from the 15N-labelled pool after start of denitrification with open chamber when production of

15N–labelled N2 and N2O would start at constant rates, leading to accumulation of 15N-labelled gases and thus to build-up of concentration gradients to the surface and to the subsoil. Concentration trends following chamber closure are shown as dotted lines.

Figure 2b: Time course of relative fluxes of N2 and N2O evolved from the 15N-labelled pool after start of denitrification with open chamber showing increasing surface and subsoil fluxes while the storage flux decreases until steady state is reached. Trends of fluxes following chamber closure are shown as dotted lines.

RC 2.15 Page 6 - Figure 2b - What is the origin of the oscillation in the flux data Page 7

AC 2.15: Response: The oscillation is a numerical artefact, that affected only the simulation of the steady state dynamics. This problem is now solved Changes: We will replace the figure with a the results of the updated model

RC 2.16 Line 1- is the chamber here fully/homogeneously mixed? AC 2.16 Response: Yes, the model assumes a fully mixed chamber. Changes: we will add the information that the model assumes a homogenously mixed chamber

RC2.17 Line 18 - Is the atmosphere multi-layer? This isn't clear AC 2.17 Response /changes: We will explain this in the description of the model

RC2.18: Line 22 (and elsewhere) - NO3 is often used, but are there any chemical or biological processes modelled that convert NO3 to other species? If not then perhaps its best to clarify that gases are produced independent of NO3 transformation. AC2.18 Response: In the methods we explain that we assume a simplified process dynamics where in terms of N transformation only nitrate reduction by denitrification occurs with N2 and N2O as emitted products. Our estimation of subsoil diffusion is based on the assumption that N2 and N2O production stay constant long enough to reach steady state before chamber closure and also do not change during chamber closure. Other

nitrate transformations, e.g., microbial immobilisation, plant uptake or leaching, would only be relevant for our approach if they cause a rapid change in N2 and N2O production so that near steady state fluxes are not reached. To exclude a discussion on the potential impact of numerous pathways of nitrate transformations we will clarify in the methods that we address N2 and N2O production from 15N labelled nitrate pool via canonical (i.e. heterorophic bacterial ) denitrification and that we assume relative constant rates so that near steady state is established. In the discussion we will briefly mention other nitrate pathways and possible impact on our results. Changes: We will add statements mentioned above in the methods and discussion sections.

RC2.19 Line 10 - Is production constant with depth over the length of the collar? Page 11

AC 2.19 Response: yes Changes: we will add that production rates are assumed constant over the length of the cylinder

RC 2.20 - Line 11 - Clarify that these initial results are from the bottom open scenario

AC 2.20 Response: We are sorry that we did not make this clear enough Changes: We will update Table 1 to include all scenarios used in the manuscript. Moreover we will refer to these scenarios in the caption of all figures and tables were scenarios results are shown.

---

## Author Response (AR1)

**General response**

We are grateful to the Editors for considering our manuscript. We thank the reviewers for their constructive comments (*repeated in italics*), which revealed several gaps and options for improvement. These were the basis for substantial improvement of the manuscript. In the following we answer all comments and declare how we changed the manuscript to fulfil the comments.

To fulfil some of the requests, we had to repeat some of the modelling scenarios and add new ones. Doing that we realized that the empirical equation to calculated diffusivity was not the most suitable one to determine Ds at high soil moisture content. To check whether this would have significant impact on our results and conclusions we repeated all simulations using the most suitable equation to calculate Ds (Millington, 1959), finding only minor changes in the results. But nevertheless we replaced all model results with the new simulation results. Moreover, we extended the discussion to cover uncertainty of our approach more detailed. Finally we deleted Figure 3a because we realized that we did not address it in the text and found it to be dispensable.

[revised manuscript text omitted]

*Dear Associate Editor,*

*The manuscript in consideration sought to elucidate whether the field application of the 15N gas flux method underestimates denitrification rates and evaluate the possible reasons by using soil diffusion modelling. The topic is of particular interest to terrestrial biogeochemists attempting to constrain N cycling pro- cesses since denitrification is notoriously difficult to measure under field conditions. The authors significantly advance knowledge in this field of research by providing a first proof of denitrification rates underestimation due to subsoil diffusion and storage of denitrification products as stipulated previously by research published in this jour- nal.*

*RC1.1*

*Even though the authors provide strong indications that subsoil diffusion is indeed occurring during field application of the 15N gas flux method I am not convinced of the practical applicability of soil diffusion modelling for correcting this discrepancy. The significant difference between measured and modelled results suggests there is too many unknown factors (e.g. spatial variability of diffusivity) and that further assump tions (beyond the homogenous soil labelling of the 15N gas flux method) need to be introduced to model surface flux and subsoil diffusion and storage.*

*I am wondering if the soil diffusivity assumptions (homogenous soil pore structure and water content, absence of stones, roots etc and constancy of diffusion and production rates) are actually introducing more bias than practical improvements to the traditional chamber method (e.g. depth of labelling, size of chamber, open or closed bottom and closure time).*

AC1.1

Response: We agree that modelling of subsoil fluxes is associated with a variety of uncertainties which we explained in before in P 19 L 20-31:

"The general agreement between measured and modelled increase in surface flux after closing the cylinder bottom is a first proof of our concept to quantify denitrification rates using surface fluxes and modelling. Reasons for the observed deviations between experimental and model results can be manifold, e.g., imperfect estimate of Ds by the empirical model (Millington & Quirk), spatial variability of diffusivity (Kuhne et al., 2012; Lange et al., 2009; Maier et al., 2017; Maier and Schack-Kirchner, 2014; Marrero, 1972) within the 10 cm layers for which Ds was determined, spatial variability of denitrification rates (Groffman et al., 2009), production of $^{15}N_2$ and $^{15}N_2O$ from possibly leached $^{15}NO_3^-$ below the confined soil cores, and a possible shift in denitrification rates during the 6 hours between the two experiments with bottom open and bottom closed. A quantitative evaluation of the model by $^{15}N$ gas flux experiments would be quite challenging since it would mean to assess all aforementioned uncertain factors and to include heterogeneity in the modelling. Future attempts are therefore necessary to improve model evaluation and check how our approach will perform under heterogenic conditions. But despite these uncertainties, the general agreement of model and measurements shows that our approach leads to improved denitrification estimates. "

But even uncertain estimates of subsoil fluxes would improve the outcome of the 15N gas flux method in comparison with current practice (i.e. without taking subsoil diffusion and storage into account) as it would lower the bias in estimating denitrification rates. We can exclude that our approach could increase bias in view of the limited effect by varying diffusivity on relative subsoil diffusion and storage flux (see also AC 2.6 below). Severe overestimation of these quantities could only occur at high soil gas diffusivity, that means in dry highly porous soils . But these conditions are not relevant for our approach since denitrification is inhibited at high diffusivity.

Changes: We added Table 4 to illustrate the range in calculated bias. Moreover , we extended the discussion on uncertainties and demonstrated that an increase in bias by calculating subsoil fluxes is improbable.

*RC1.2*

*The authors suggest that previously published data could be corrected for underestimation by using their model with further parameterisation. This indeed would be something I would be very interested to see and particularly for more challenging soil types than arable land such as grasslands or forests.*

AC1.2

Response: Thank you for this suggestion. We plan to do this in follow-up studies

In the conclusions we now mention that follow-up studies are needed to obtain further model parametrization to enable correction of previous published in situ N2+N2O fluxes covering all land use types

*The manuscript is well structured and clearly written and it seems to me it is a first step towards the right direction for further improving field denitrification measurements. I therefore recommend that the manuscript is accepted for publication following a few minor corrections and clarifications detailed below:*

*Minor comments:*

*RC1.3*

*1. P1 Lines 18: End the sentence after total production and start a new one after it.*

AC1.3 Changes: done as suggested

*RC1.4*

*2. P3 L9 and throughout: Please correct spelling of the word labelled throughout the manuscript.*

AC 1.4 Changes: done as suggested

*RC1.5*

*3. P3 L25&26: In Sgouridis et al. 2016 the labelled nitrate was applied via injections to the soil volume. Please correct the reference and replace with one that surface application was used.*

AC1.5 Changes: done as suggested

*RC1.6*

*4. P4 L16: Was steady state within the first 6 hours after the label application also measured or just modelled? In the next sentence the assumption stated is that gas production starts at constant rates after the label application. Is it therefore necessary to first establish steady state before applying the model?*

AC1.5

Response: Indeed the model can only yield correct values if fluxes are steady state at chamber closing. Hence, surface flux data collected immediately after labelling where activity and fluxes dramatically change over time could not be corrected exactly. Moreover, we did not determine steady state experimentally. However, measurements were conducted 5 days after labelling (section 2.3). Therefore, since this is 20 times the modelled steady state time, we can expect that near steady state was reached, even though we could not check this. But we realize that incomplete steady state, due to changes in activity and/or diffusivity, e.g., following precipitation, could be an issue. We will include this in the discussion.

Changes: In the discussion we added a statement that further uncertainty could arise from incomplete steady state , e.g., following precipitation and thus decreasing diffusivity, increasing moisture and change in the labelled volume (P23, L22-24), and that this effect should be evaluated in follow-up studies (P24, L32f).

*RC1.6.*

*P10 L22: Reference is repeated twice.*

AC1.6 Changes: corrected

*RC1.7*

*P11 L26: It would have been useful if measurements with or without closed bottom cylinder and varying labelling depths and lengths of cylinders were also taken. This could have shown whether the model predictions are true and if there is a significant difference in surface fluxes to justify the use of the model. Perhaps a combination of lower labelling depth, deeper cylinder and larger chamber would result in insignificant subsoil diffusion losses.*

AC1.7

Response: This is indeed planned for a follow up study

Changes: In the discussion we explained that the applicability of our approach for varying depth of labelling should be checked by future studies (P25, L1).

*RC1.8*

*P19 L2: I agree that it would be a lot easier to apply the model under laboratory closed system conditions. However, pore space/headspace equilibration is relatively easier to achieve than attempting the soil diffusion modelling. The real challenge for the future application of the model would be to apply it under field conditions in more challenging soil types.*

AC1.8

Response: We agree. Closed system was mentioned because there is also some effect that had not been taken into account until now. The challenge for the future application of the model would be to apply it under field conditions in more challenging soil types as was already addressed in the following section in page 19, where we discuss the potential and limitation of our approach for field studies.

Changes: In the discussion we deepened the discussion on the challenges to further evaluate our approach. In the conclusions we address the need to cover all soil types.
*Overall the paper is quite relevant to researchers who have used or are planning to use the 15N labelling method to quantify identification rates in-situ. The researchers convincingly show through modelling and field data that the impact of subsoil diffusion and storage fluxes have a significant impact on the estimated denitrification rates and thus have likely caused under reporting in the current literature.*

*General comments*

*R2.1*

*- My main reservation is regarding the applicability of the modelling more broadly for correction of field results. Parameters like diffusivity are notoriously difficult to estimate in the field, and therefore the discrepancy between model and measured as reported here may never be reconcilable.*

AC2.1

Response: This was also addressed by reviewer 1 (RC1.1). We agree that exact prediction of subsoil diffusion is difficult, but determination of Ds in the subsoil is feasible with reasonable effort and can also be modelled based on bulk density and soil moisture. This will lead to Ds estimates accurate enough for subsoil diffusion modelling (see also AC2.6 below).

Changes: In the results we added additional scenarios to illustrate the impact from uncertain estimates of soil moisture and thus diffusivity (Table 3).

In the discussion we added a section (P25, L 15f) to explain that taking into account maximum uncertainty in subsoil diffusion modelling we demonstrate that worst case scenarios would still improve estimates compared to previous practice. Here we also explained that small scale heterogeneity in Ds had little or moderate effect on simulate subsoil diffusion.

*R2.2*

*- While i appreciate the difficulty of including water phase gas transport in a model, especially one with such a complicated isotopologue structure, I feel it should at least be discussed in the paper as another important factor. It would both contribute to pore space storage as well as isotopic fractionation although the latter may not be important given the label strength.*

AC2.2

Response: We agree, especially in case of N2O due its high solubility in water, there would be some storage which would have some impact on the change of fluxes following chamber closing due to the slow diffusion in the water phase. This should be tested in follow up studies.

Changes: In the discussion we now mention that water phase transport was not yet taken into account, but might have some relevance due low diffusivity in the water phase. We also mention that this effect would be largest for N2O due to its high solubility in water and will add references were the impact of water phase gas transport is addressed with respect to CO2 flux modelling. We

also mention that water phase dynamics might be another explanation for the deviations between the N2O/(N2+N2O) ratios determined with bottom open and bottom closed (P23 , L 26-32).

*RC2.3*

*- The model results are somewhat dense and difficult to digest - my concern is that someone who is not a modeller/gas diffusion specialist would get lost in the current brief narrative. Suggest being more verbose but for the benefit on enhanced clarity.*

AC2.3

Response: Wwe are sorry for this.

Changes: The modelling results were restructured and extended.

*RC2.4*

*- Section 3.1.2 requires significantly more explanation. I would have expected to see a more normal flux calculation as a proxy for production as is done with CO2 or CH4, however the fitting approach is applied here. Why did the authors not use a linear or exponential flux model as is commonly used for other gases. What do the parameters alpha and delta signify or what is their physical manifestation - are they related to chamber volume and surface area, cylinder depth, etc? Is this approach/equation commonly applied outside of this paper?*

AC 2.4 Response: we are sorry for this. We will supply adequate explanations.

Changes: We added additional explanations to section 3.1.2 to clarify that the empirical equation was used to generate data for our field comparison and also as an example how denitrification rates could be obtained in future without the need to run the 3 D model for each experiment or sampling event.

*- Discussion and conclusions*

*RC2.6*

*- If the modelling approach cannot be applied quite yet to correct the values, perhaps there should be a small table or histogram or similar of "likely errors" that may have been incurred in past experiments using this method. This would at least allow the community to make an educated guess on how far off our current estimates are from reality (and may allow some reconciliation across methods as well).*

AC 2.6

Response: We agree that this is would be useful..

Changes: We generated Table 3 based on the scenarios we already included in the manuscript, and added additional new scenarios. Table 3 shows % underestimation for our micro-plot and chamber geometry in dependence of soil moisture (and thus Ds) and chamber deployment time. We discussed how results would change with differing depth in labelling or size of the micro-plots.

*RC2.7*

*- Overall the flow of the paper could be improved, this is partly due to sections with poor sentence structure or run-on thoughts mostly in the introduction and discussion portion of the paper.*

AC2.7

Response: We are sorry for this. We evaluated and improve the flow of the paper and checked sentence structures.

Changes: We worked on the flow and sentence structure as requested. In the introduction, long sentences were shortened and some sections were restructured.

*Specific Comments*

*Page 2*

*RC2.8*

*- Line 9 - "to measure" should be "in measuring" or similar.*

AC2.8 Response/changes: done as suggested

*RC2.9*

*- Line 14 - gastight should be gas tight unless this is a brand of container*

AC2.9 Response/changes: done as suggested

*RC2.10*

*- Line 25 - suggest inserting several sentences explaining to the reader why in-situ measurements are important. Is there literature to cite comparing in-situ to lab incubations or similar?*

AC 2-10

Response: Denitrification is complexly controlled by interaction of labile C, abundance and community structure of denitrifiers, pore structure, soil and root respiration, mineral N dynamics. Hence, it is difficult to keep conditions in the lab identical to the field where some conditions dynamically change due to climatic factors but especially due to the activity of plants. Field measurements are therefore needed for reliable determination of denitrification in ecosystems.

There are numerous lab studies but few field measurements. To the best of our knowledge respective comparisons are still missing for unsaturated soils. We conducted such a comparison only for denitrification in shallow groundwater (Well, R., et al. (2003). "Comparison of field and laboratory measurement of denitrification and N2O production in the saturated zone of hydromorphic soils." Soil Biology & Biochemistry 35(6): 783-799.

Changes: The explanation for the need of field studies was included in the introduction.

*Page 3*

*RC 2.11*

*- Line 8 - some more detail around why we don't just measure these parameters instead of modelling them*

AC 2.11

Response: we are not sure if we understand this question correctly. Line 8 f reads: "Modelling diffusion of 15N2 + 15N2O produced in 15N-lablelled surface soil could be used to estimate its accumulation in pore space and diffusive loss to the subsoil and thus to quantify denitrification from the sum of surface flux, subsoil flux and storage within the 15N-labelled soil volume." By modelling diffusion we mean: modelling the diffusive flux. But the diffusive flux can not be measured directly. But unfortunately we failed to mention that this approach would include measurement of surface flux.

Changes: We reformulated this sentence as: "Modelling diffusive fluxes of $^{15}N_2$ + $^{15}N_2O$ produced in $^{15}N$-labelled surface soil based on measured surface flux and diffusivity could be used to estimate its accumulation in pore space and diffusive loss to the subsoil. This could be used to quantify denitrification from the sum of surface flux, subsoil flux and storage within the $^{15}N$-labelled soil volume. " (P3, L 18f)

*RC2.12*

*- Line 14 - change amount to concentration*

AC 2.12 done as suggested

*RC2.13*

*- Lines 18-29 - consider separating into bullets.*

AC 2.13 done as suggested

*RC2.14*

*-Consider annotating figures with some of the details contained in lines 18-29*

AC 2.14

Response: done as suggested:

Changes: Modified caption of Fig. 2a,b:

Figure 2a: Increase in pore space concentrations of N2 evolved from the 15N-labelled pool after start of denitrification with open chamber when **production of 15N–labelled N2 and N2O would start at constant rates, leading to accumulation of 15N-labelled gases and thus to build-up of concentration gradients to the surface and to the subsoil.** Concentration trends following chamber closure are shown as dotted lines.

Figure 2b: Time course of relative fluxes of N2 and N2O evolved from the 15N-labelled pool after start of denitrification with open chamber **showing increasing surface and subsoil fluxes while the storage flux decreases until steady state is reached.** Trends of fluxes following chamber closure are shown as dotted lines.

*RC 2.15*

*Page 6 - Figure 2b*

*- What is the origin of the oscillation in the flux data Page 7*

AC 2.15:

Response: The oscillation is a numerical artefact, that affected only the simulation of the steady state dynamics. This problem is now solved, i.e. numerical oscillation is now much smaller.

Changes: We replaced the figure with a the results of the updated model

*RC 2.16*

*Line 1- is the chamber here fully/homogeneously mixed?*

AC 2.16

Response: Yes, the model assumes a fully mixed chamber.

Changes: we will added the information that the model assumes a homogenously mixed chamber

*RC2.17*

*Line 18 - Is the atmosphere multi-layer? This isn't clear*

AC 2.17 Response /changes:  We added the explanation that the free atmosphere was simulated as a well mixed single layer.

*RC2.18:*

*Line 22 (and elsewhere) - NO3 is often used, but are there any chemical or biological processes modelled that convert NO3 to other species? If not then perhaps its best to clarify that gases are produced independent of NO3 transformation.*

AC2.18

Response: In the methods we explain that we assume a simplified process dynamics where in terms of N transformation only nitrate reduction by denitrification occurs with N2 and N2O as emitted products. Our estimation of subsoil diffusion is based on the assumption that N2 and N2O production stay constant long enough to reach steady state before chamber closure and also do not change during chamber closure. Other nitrate transformations, e.g., microbial immobilisation, plant uptake or leaching, would only be relevant for our approach if they cause a rapid change in N2 and N2O

production so that near steady state fluxes are not reached. To exclude a discussion on the potential impact of numerous pathways of nitrate transformations we will clarify in the methods that we address N2 and N2O production from 15N labelled nitrate pool via microbial denitrification and that we assume relative constant rates so that near steady state is established.

Changes: statements mentioned above were added in the methods and discussion sections

*Page 8*

*RC2.19*

*Line 10 - Is production constant with depth over the length of the collar? Page 11*

AC 2.19

Response: yes

Changes: we added that production rates are assumed constant within in the labelled soil

*RC 2.20*

*- Line 11 - Clarify that these initial results are from the bottom open scenario*

AC 2.20

Response: We are sorry that we did not make this clear enough

Changes: We updated Table 1 to include all scenarios used in the manuscript. Moreover we referred to these scenarios in the caption of the figures and tables were scenarios results are shown.

**Underestimation of denitrification rates from field application of the $^{15}$N gas flux method and its correction by gas diffusion modelling**

Reinhard Well[1], Martin Maier[2], Dominika Lewicka-Szczebak[1], Jan-Reent Köster[1], Nicolas Ruoss[1]

[1]Thünen Institute, Climate-Smart Agriculture, Braunschweig, Germany
[2]Forest Research Institute Baden-Württemberg, Dep. Soil and Environment, Freiburg, Germany

*Correspondence to*: Reinhard Well (reinhard.well@thuenen.de)

**Supplement**

Table S1: Soil data (WFPS = water-filled pores space; means± standard deviation of four replicate micro-plots)

| Depth of sample | WFPS | NO$_3^-$ | NH$_4^+$ | $^{15}$N atom fraction of NO$_3^-$ | Bulk density |
|---|---|---|---|---|---|
| | % | mg N kg$^{-1}$ | mg N kg$^{-1}$ | | g cm-3 |
| **0-10 cm** | 71.8±2.6 | 16.6±1.9 | 1.76±1.05 | 0.092±0.014 | 1.48 |
| **10-20 cm** | 61.5±2.4 | 14.4±2.5 | 0.81±0.32 | 0.150±0.045 | 1.54 |
| **20-30 c.** | 60.0±1.5 | 16.6±4.1 | 0.70±0.18 | 0.201±0.045 | 1.48 |
| **0-30 cm (average)** | 64.4±1.7 | 15.9±2.5 | 1.1±0.4 | 0.148±0.030 | 1.50 |

**Table S2: Field fluxes of pool-derived N₂, N₂O and N₂+N₂O, residual fraction of N₂O remaining after N₂O reduction to N₂ ($r_{N2O}$) and $^{15}N$ enrichment of the $^{15}N$-labelled N pool producing N₂O ($a_{p\_N2O}$) with bottom open and bottom closed (individual replicates and mean values ± standard deviation). Unequal uppercase letter indicate significant (P<0.05) differences between mean values with bottom open and bottom closed.**

| ID | N₂ flux | N₂O flux | N₂+N₂O flux | $r_{N2O}$ | $a_{p\_N2O}$ |
|---|---|---|---|---|---|
| | g N ha⁻¹ d⁻¹ | g N ha⁻¹ d⁻¹ | g N ha⁻¹ d⁻¹ | | |
| Cylinder 1 / bottom open | 286.3 | 62.1 | 348.4 | 0.178 | 0.126 |
| Cylinder 2 / bottom open | 436.0 | 73.9 | 509.9 | 0.145 | 0.194 |
| Cylinder 3/ bottom open | 763.9 | 237.6 | 1001.4 | 0.237 | 0.113 |
| Cylinder 4 / bottom open | 488.2 | 9.6 | 497.8 | 0.019 | 0.174 |
| average, bottom open | 493.6ᵃ±199.5 | 95.8ᵃ±98.5 | 589.4ᵃ±284.3 | 0.145ᵃ±0.092 | 0.152ᵃ±0.038 |
| | | | | | |
| Cylinder 1 / bottom closed | 349.9 | 139.4 | 489.3 | 0.285 | 0.120 |
| Cylinder 2 / bottom closed | 776.2 | 30.3 | 806.5 | 0.038 | 0.202 |
| Cylinder 3/ bottom closed | 1150.7 | 170.7 | 1321.3 | 0.129 | 0.121 |
| Cylinder 4 / bottom closed | 540.0 | 62.5 | 602.5 | 0.104 | 0.177 |
| average, bottom closed | 704.2ᵃ±345.0 | 100.7ᵃ±65.4 | 804.9ᵇ±368.5 | 0.139ᵃ±0.105 | 0.155ᵃ±0.041 |

[Figure]

[Figure]

**Figure S1: Simulation of concentrations (colours, ppm) and fluxes (arrows) with open chamber at steady state.**

[Figure]

[Figure]

**Figure S2: Simulation of concentrations (colours, ppm) and fluxes (arrows) 5 hours after chamber closure.**

[Figure]

[Figure]

¶
¶
,
¶

**Figure S3 Relative fluxes of N₂ isotopologues ($^{14}N^{14}N$, $^{15}N^{14}N$, $^{15}N^{15}N$) following chamber closing.**

---

## Author Response (AR2)

Dear Dr. Trebs,

thank you very much for final acceptance of our manuscript.

We thank the reviewers again for their constructive work and final check of our revised manuscript.

We submitted all files needed for final production. Since there was no request for changes in the final submission we kept the manuscript unchanged except for inserting the section on authors contributions.

Best regards,

Reinhard Well